# Differential dopaminergic modulation of spontaneous cortico–subthalamic activity in Parkinson's disease

Abhinav Sharma[1]*, Diego Vidaurre[2,3], Jan Vesper[4], Alfons Schnitzler[1,5], Esther Florin[1]*

[1]Institute of Clinical Neuroscience and Medical Psychology, Medical Faculty, Heinrich-Heine University Düsseldorf, Düsseldorf, Germany; [2]Department of Psychiatry, University of Oxford, Oxford, United Kingdom; [3]Department of Clinical Health, Aarhus University, Aarhus, Denmark; [4]Department of Neurosurgery, University Hospital Düsseldorf, Düsseldorf, Germany; [5]Department of Neurology, Center for Movement Disorders and Neuromodulation, Medical Faculty, Heinrich-Heine University Düsseldorf, Düsseldorf, Germany

**Abstract** Pathological oscillations including elevated beta activity in the subthalamic nucleus (STN) and between STN and cortical areas are a hallmark of neural activity in Parkinson's disease (PD). Oscillations also play an important role in normal physiological processes and serve distinct functional roles at different points in time. We characterised the effect of dopaminergic medication on oscillatory whole-brain networks in PD in a time-resolved manner by employing a hidden Markov model on combined STN local field potentials and magnetoencephalography (MEG) recordings from 17 PD patients. Dopaminergic medication led to coherence within the medial and orbitofrontal cortex in the delta/theta frequency range. This is in line with known side effects of dopamine treatment such as deteriorated executive functions in PD. In addition, dopamine caused the beta band activity to switch from an STN-mediated motor network to a frontoparietal-mediated one. In contrast, dopamine did not modify local STN–STN coherence in PD. STN–STN synchrony emerged both on and off medication. By providing electrophysiological evidence for the differential effects of dopaminergic medication on the discovered networks, our findings open further avenues for electrical and pharmacological interventions in PD.

*For correspondence:
Abhinav.Sharma@med.uni-duesseldorf.de (AS);
esther.florin@hhu.de (EF)

## Introduction

Oscillatory activity serves crucial cognitive roles in the brain (*Akam and Kullmann, 2010*; *Akam and Kullmann, 2014*), and alterations of oscillatory activity have been linked to neurological and psychiatric diseases (*Schnitzler and Gross, 2005*). Different large-scale brain networks operate with their own oscillatory fingerprint and carry out specific functions (*Keitel and Gross, 2016*; *Mellem et al., 2017*; *Vidaurre et al., 2018b*). Given the dynamics of cognition, different brain networks need to be recruited and deployed flexibly. Hence, the duration for which a network is active, its overall temporal presence, and even the interval between the different activations of a specific network might provide a unique window to understanding brain functions. Crucially, alterations of these temporal properties or networks might be related to neurological disorders.

In Parkinson's disease (PD), beta oscillations within the subthalamic nucleus (STN) and motor cortex (13–30 Hz) correlate with the motor symptoms of PD (*Marreiros et al., 2013*; *van Wijk et al., 2016*; *West et al., 2018*). Beta oscillations also play a critical role in communication in a healthy brain (*Engel and Fries, 2010*). (For the purposes of our paper, we refer to oscillatory activity or oscillations as recurrent but transient frequency-specific patterns of network activity, even though the

underlying patterns can be composed of either sustained rhythmic activity, neural bursting, or both [*Quinn et al., 2019*]. Disambiguating the exact nature of these patterns is, however, beyond the scope of this work.) At the cellular level, loss of nigral dopamine neurons in PD leads to widespread changes in brain networks, to varying degrees across different patients. Dopamine loss is managed in patients via dopaminergic medication. Dopamine is a widespread neuromodulator in the brain (*Gershman and Uchida, 2019*), raising the question of whether each medication-induced change restores physiological oscillatory networks. In particular, dopaminergic medication is known to produce cognitive side effects in PD patients (*Voon et al., 2009*). According to the dopamine overdose hypothesis, a reason for these effects is the presence of excess dopamine in brain regions not affected in PD (*MacDonald et al., 2011*; *MacDonald and Monchi, 2011*). Previous task-based and neuroimaging studies in PD demonstrated frontal cognitive impairment due to dopaminergic medication (*Cools et al., 2002*; *Ray and Strafella, 2010*; *MacDonald et al., 2011*).

Using resting-state whole-brain MEG analysis, network changes related to both motor and non-motor symptoms of PD have been described (*Olde Dubbelink et al., 2013a*; *Olde Dubbelink et al., 2013b*). However, these studies could not account for simultaneous STN–STN or cortico–STN activity affecting these networks, which would require combined MEG/electroencephalogram (EEG)–LFP recordings (*Litvak et al., 2021*). Such recordings are possible during the implantation of deep brain stimulation (DBS) electrodes, an accepted treatment in the later stages of PD (*Volkmann et al., 2004*; *Deuschl et al., 2006*; *Kleiner-Fisman et al., 2006*). Combined MEG–LFP studies in PD involving dopaminergic intervention report changes in beta and alpha band connectivity between specific cortical regions and the STN (*Litvak et al., 2011*; *Hirschmann et al., 2013*; *Oswal et al., 2016*). Decreased cortico–STN coherence under dopaminergic medication (ON) correlates with improved motor functions in PD (*George et al., 2013*). STN–STN intra-hemispheric oscillations positively correlate to motor symptom severity in PD without dopaminergic medication (OFF), whereas dopamine-dependent nonlinear phase relationships exist between inter-hemispheric STN–STN activity (*West et al., 2016*). Crucially, previous studies could not rule out the influence of cortico–STN connectivity on these inter-hemispheric STN–STN interactions.

To further characterise the differential effects of dopaminergic medication and delineate pathological versus physiological-relevant spectral connectivity in PD, we study PD brain activity via a hidden Markov model (HMM), a data-driven learning algorithm (*Vidaurre et al., 2016*; *Vidaurre et al., 2018b*). Due to the importance of cortico–subcortical interactions in PD, we investigated these interactions with combined spontaneous whole-brain magnetoencephalography (MEG) and STN local field potentials (LFPs) recordings from PD patients. We study whole-brain connectivity including the STN using spectral coherence as a proxy for communication based on the communication through coherence hypothesis (*Fries, 2005*; *Fries, 2015*). This will allow us to delineate differences in communication OFF and ON medication. Furthermore, we extended previous work that was limited to investigating communication between specific pairs of brain areas (*Litvak et al., 2011*; *George et al., 2013*; *Hirschmann et al., 2013*). Moreover, we identified the temporal properties of the networks both ON and OFF medication. The temporal properties provide an encompassing view of network alterations in PD and the effect of dopamine on these networks.

We found that cortico–cortical, cortico–STN, and STN–STN networks were differentially modulated by dopaminergic medication. For the cortico–cortical network, medication led to additional connections that can be linked to the side effects of dopamine. At the same time, dopamine changed the cortico–STN network towards a pattern more closely resembling physiological connectivity as reported in the PD literature. Within the third network, dopamine only had an influence on local STN–STN coherence. These results provide novel information on the oscillatory network connectivity occurring in PD and the differential changes caused by dopaminergic intervention. These whole-brain networks, along with their electrophysiological signatures, open up new potential targets for both electric and pharmacological interventions in PD.

## Results

Under resting-state conditions in PD patients, we simultaneously recorded whole-brain MEG activity with LFPs from the STN using directional electrodes implanted for DBS. Using an HMM, we identified recurrent patterns of transient network connectivity between the cortex and the STN, which we henceforth refer to as an 'HMM state'. In comparison to classic sliding window analysis, an HMM

solution can be thought of as a data-driven estimation of time windows of variable length (within which a particular HMM state was active): once we know the time windows when a particular state is active, we compute coherence between different pairs of regions for each of these recurrent states. Each HMM state itself is a multidimensional, time-delay embedded (TDE) covariance matrix across the whole brain, containing information about cross-regional coherence and power in the frequency domain. Additionally, the temporal evolution of the HMM states was determined. The PD data were acquired under medication (L-DOPA) OFF and ON conditions, which allowed us to delineate the physiological versus pathological spatio-spectral and temporal changes observed in PD. To allow the system to dynamically evolve, we use time delay embedding. Theoretically, delay embedding can reveal the state space of the underlying dynamical system (*Packard et al., 1980*). Thus, by delay-embedding PD time series OFF and ON medication, we uncover the differential effects of a neurotransmitter such as dopamine on underlying whole-brain connectivity. OFF medication, patients had on average a Unified Parkinson's Disease Rating Scale (UPDRS) part III of 29.24 ± 10.74. This was reduced by L-DOPA (176.5 ± 56.2 mg) to 19.47 ± 8.52, indicating an improvement in motor symptoms.

## Spontaneous brain activity in PD can be resolved into distinct states

Using an HMM, we delineated cortico–subthalamic spectral changes from both global source-level cortical interactions as well as local STN–STN interactions. Three of the six HMM states could be attributed to physiologically interpretable connectivity patterns. We could not interpret the other three states within the current physiological frameworks both OFF and ON medication and they are therefore not considered in the following (see *Figure 2—figure supplement 1*). The connectivity between different brain regions for each state was visualised for the frequency modes shown in *Figure 1*. *Figures 2–4* show the connectivity patterns for the three physiologically meaningful states in both the OFF (top row) and ON medication condition (bottom row). We refer to the state obtained in *Figure 2* as the cortico–cortical state (*Ctx–Ctx*). This state was characterised mostly by local coherence within segregated networks OFF medication in the alpha and beta band. In contrast, there was

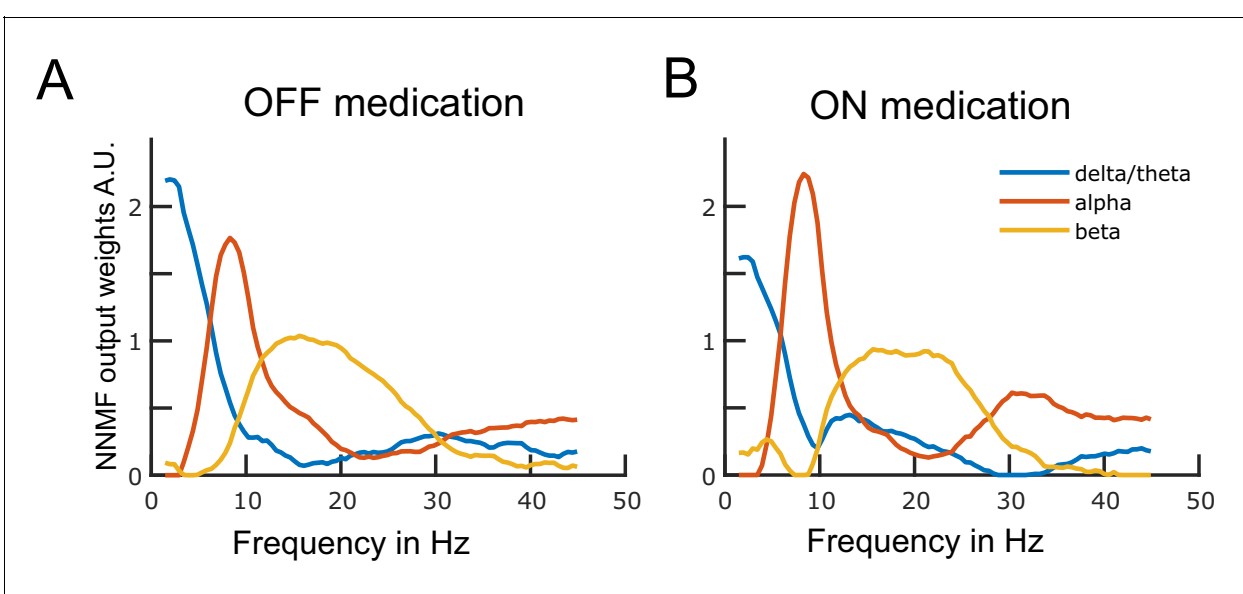

**Figure 1.** Data-driven frequency modes. Each plotted curve shows a different spectral band. The x-axis represents frequency in Hz and the y-axis represents the weights obtained from the non-negative matrix factorisation (NNMF) in arbitrary units. The NNMF weights are like regression coefficients. The frequency resolution of the modes is 0.5 Hz. Panels A and B show the OFF and ON medication frequency modes, respectively. Source data are provided as *Figure 1—source data 1–2*.

The online version of this article includes the following source data for figure 1:

**Source data 1.** Source data of *Figure 1a*.
**Source data 2.** Source data of *Figure 1b*.

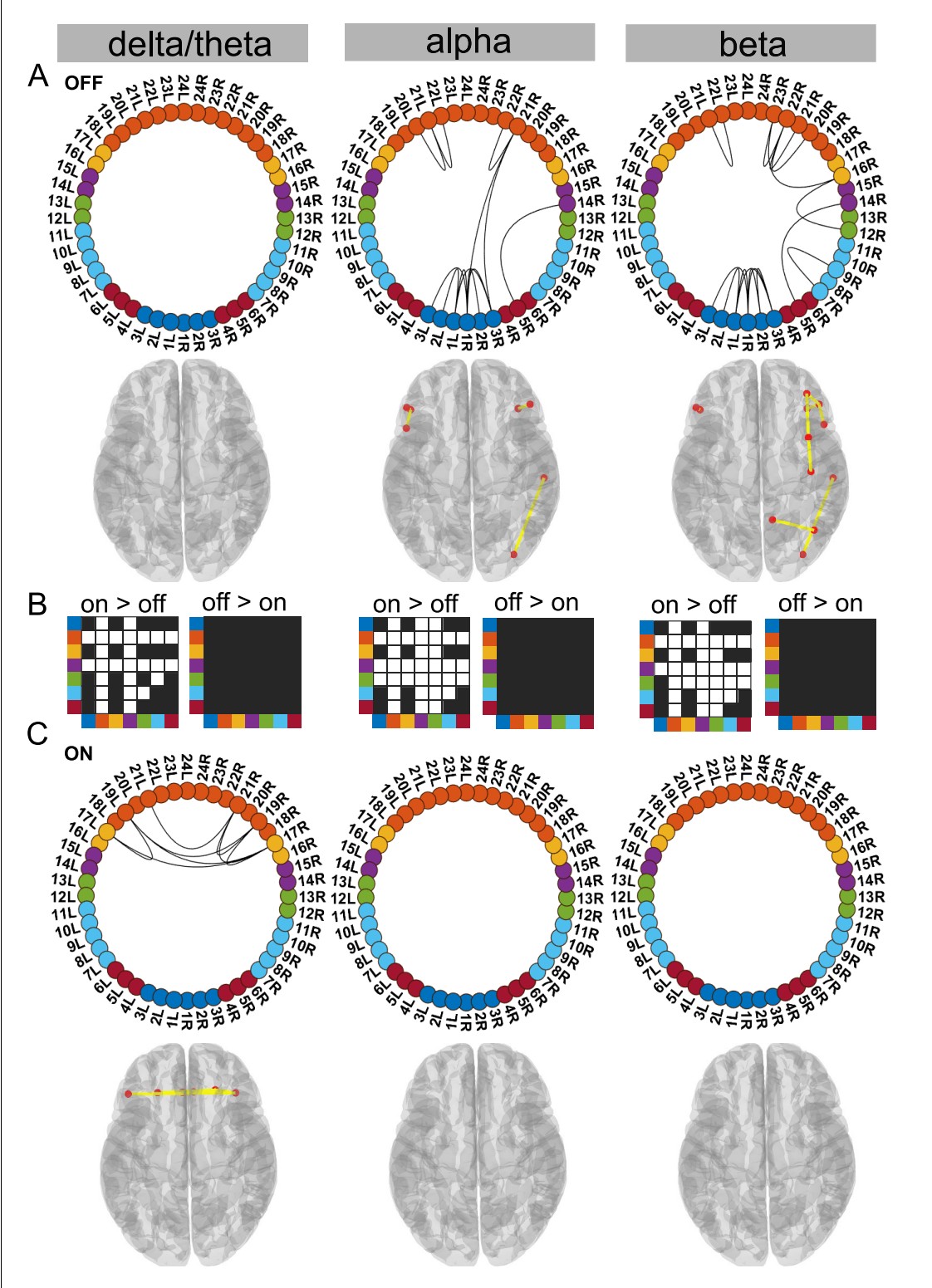

**Figure 2.** Cortico–cortical state. The cortico–cortical state was characterised by a significant increase in coherence ON compared to OFF medication (see panel **B**). Due to this, no connections within the alpha and beta band ON medication were significantly higher than the mean (panel **C**). However, in the delta band, ON medication medial prefrontal–orbitofrontal connectivity emerged. (**A and C**) Each node in the circular graph represents a brain region based on the Mindboggle atlas. The regions from the atlas are listed in *Table 1* along with their corresponding numbers that are used in the circular graph. The colour code in the circular graph represents a group of regions clustered according to the atlas (starting from node number 1) STN

*Figure 2 continued on next page*

*Figure 2 continued*

contacts (contacts 1, 2, 3 = right STN and contacts 4, 5, 6 = left STN), frontal, medial frontal, temporal, sensorimotor, parietal, and visual cortices. In the circular graph, only the significant connections (p<0.05; corrected for multiple comparisons, IntraMed analysis) are displayed as black curves connecting the nodes. The circles from left to right represent the delta/theta, alpha, and beta bands. Panel A shows results for OFF medication data and panel C for the ON medication condition. For every circular graph, we also show a corresponding top view of the brain with the connectivity represented by yellow lines and the red dot represents the anatomical seed vertex of the brain region. Only the cortical connections are shown. Panel **B** shows the result for inter-medication analysis (InterMed) for the cortico–cortical state. In each symmetric matrix, every row and column corresponds to a specific atlas cluster denoted by the dot colour on the side of the matrix. Each matrix entry is the result of the InterMed analysis where OFF medication connectivity between ith row and jth column was compared to the ON medication connectivity between the same connections. A cell is white if the comparison mentioned on top of the matrix (either ON >OFF or OFF >ON) was significant at a threshold of p<0.05. The connectivity maps of states 4–6 are provided in *Figure 2—figure supplement 1*. Source data are provided as *Figure 2—source data 1–3*.

The online version of this article includes the following source data and figure supplement(s) for figure 2:

**Source data 1.** Source data of *Figure 2a*.
**Source data 2.** Source data of *Figure 2b*.
**Source data 3.** Source data of *Figure 2c*.
**Figure supplement 1.** Three additional states that were found in the OFF and ON conditions.

a widespread increase in coherence across the brain from OFF to ON medication. Therefore, ON medication, the connectivity strength in the alpha and beta band was not significantly different from the mean noise level. *Figure 3* displays the second state. A large proportion of spectral connections in this state enable cortico–STN communication via spectral coherence (*Lalo et al., 2008*; *Litvak et al., 2011*; *Hirschmann et al., 2013*; *Oswal et al., 2013*; *van Wijk et al., 2016*) and thus we labelled this as the cortico–STN state (*Ctx–STN*). This state was characterised by connectivity between multiple cortical regions and the STN OFF medication, but increased specificity of cortical–STN connectivity ON medication. Finally, *Figure 4* shows the third state. Within this state, highly synchronous STN–STN spectral connectivity emerged, both OFF and ON medication and therefore we named it the STN–STN state (*STN–STN*). The spectral characteristics of this state largely remain unaffected under the influence of dopaminergic medication. In the following sections, we describe these three states in detail.

## Ctx–Ctx state is characterised by increased frontal coherence due to elevated dopamine levels

Supporting the dopamine overdose hypothesis in PD (*Kelly et al., 2009*; *MacDonald and Monchi, 2011*), we identified a delta/theta oscillatory network involving intra-hemispheric connections between the lateral and medial orbitofrontal cortex as well as the pars orbitalis. The delta/theta network emerged between the lateral and medial orbitofrontal as well as left and right pars orbitalis cortex ON medication (p<0.05, *Figure 2C* delta). On the contrary, OFF medication no significant connectivity was detected in the delta/theta band. In the alpha and beta band OFF medication there was significant connectivity within the frontal regions, STN, and to a limited extent in the posterior parietal regions (p<0.05, *Figure 2A*).

Another effect of excess dopamine was significantly increased connectivity of frontal cortex and temporal cortex both with the STN and multiple cortical regions across all frequency modes (p<0.01, *Figure 2* delta, alpha, and beta). The change in sensorimotor–STN connectivity primarily took place in the alpha band with an increased ON medication. Sensorimotor–cortical connectivity was increased ON medication across multiple cortical regions in both the alpha and beta band (p<0.01, *Figure 2* alpha and beta). However, STN–STN coherence remained unchanged OFF versus ON medication across all frequency modes.

Viewed together, the Ctx–Ctx state captured increased coherence across the cortex ON medication within the alpha and beta band. This, however, implies that ON medication, no connectivity strength was significantly higher than the mean noise level within the alpha or beta band. ON medication, significant coherence emerged in the delta/theta band primarily between different regions of the orbitofrontal cortex.

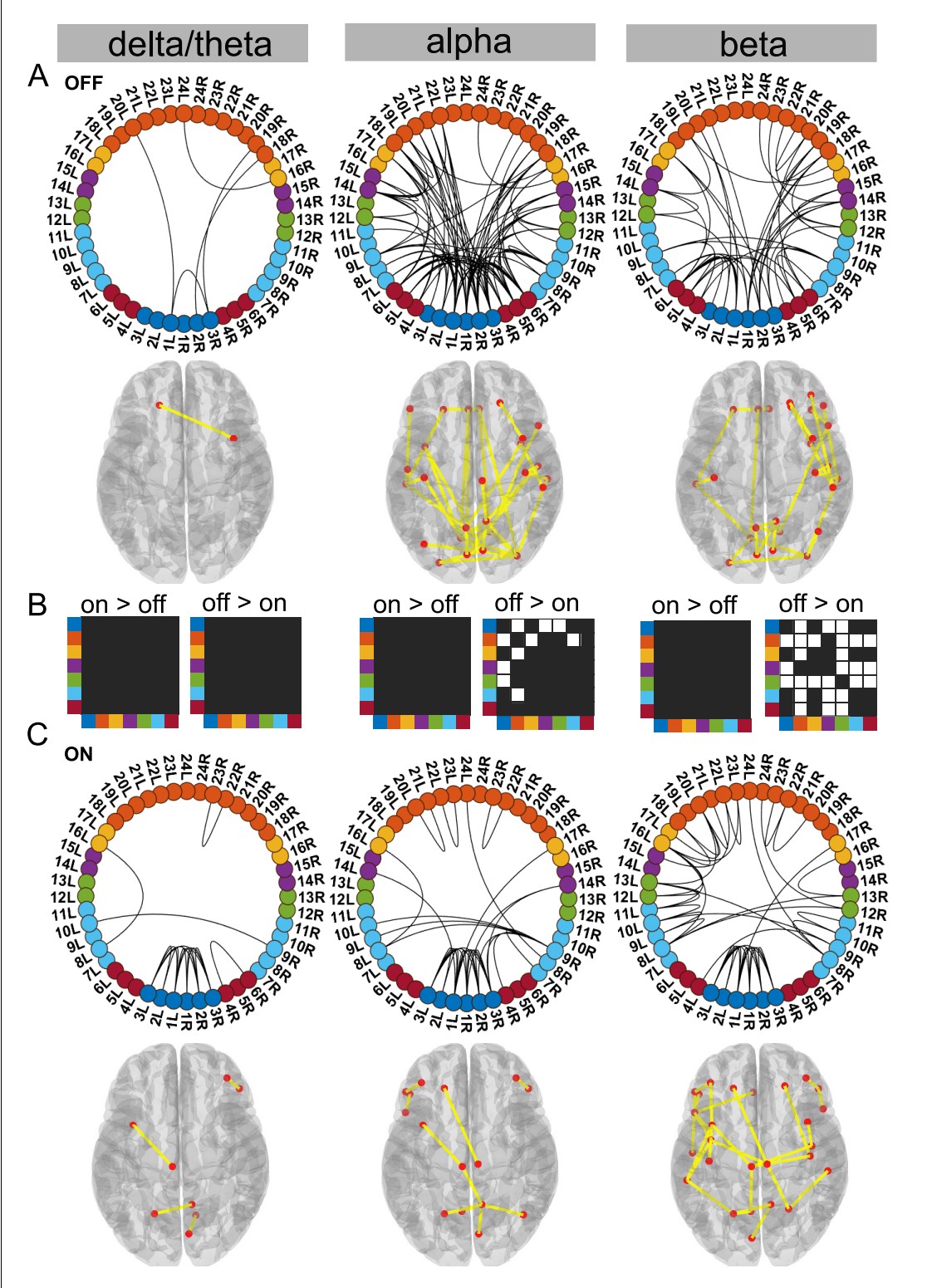

**Figure 3.** Cortico–STN state. For the general description, see the note to *Figure 2*. The cortico–STN state was characterised by preservation of spectrally selective cortico–STN connectivity ON medication. Also, ON medication, a sensorimotor–frontoparietal network emerged. Source data are provided as *Figure 3—source data 1–3*.

The online version of this article includes the following source data for figure 3:

**Source data 1.** Source data of *Figure 3a*.

**Source data 2.** Source data of *Figure 3b*.

**Source data 3.** Source data of *Figure 3c*.

## Dopaminergic medication selectively reduced connectivity in the Ctx–STN state

Our analysis revealed that the Ctx–STN state ON medication was characterised by selective cortico–STN spectral connectivity and an overall shift in cortex-wide activity towards physiologically relevant network connectivity. In particular, ON medication, connectivity between STN and cortex became more selective in the alpha and beta band. OFF medication, STN–pre-motor (sensory), STN–frontal, and STN–parietal connectivity was present (p<0.05, *Figure 3A* alpha and beta). Importantly, coherence OFF medication was significantly larger than ON medication between STN and sensorimotor, STN and temporal, and STN and frontal cortices (p<0.05 for all connections, *Figure 3B* alpha and beta). Furthermore, ON medication, in the alpha band only the connectivity between temporal, parietal, and medial orbitofrontal cortical regions and the STN was preserved (p<0.05, *Figure 3C* alpha). Finally, ON medication, a sensorimotor–frontoparietal network emerged (p<0.05, *Figure 3C* beta), where sensorimotor, medial prefrontal, frontal, and parietal regions were no longer connected to the STN, but instead directly communicated with each other in the beta band. Hence, there was a transition from STN-mediated sensorimotor connectivity to the cortex OFF medication to a more direct cortico–cortical connectivity ON medication.

Simultaneously to STN–cortico and cortico–cortical, STN–STN connectivity changed. In the ON condition, STN–STN connectivity was significantly different from the mean noise level across all three frequency modes (p<0.05, *Figure 3C*). But on the other hand, there was no significant change in the STN–STN connectivity OFF versus ON medication (p=0.21 delta/theta; p=0.25 alpha; p=0.10 beta; *Figure 3B*).

To summarise, coherence decreased ON medication across a wide range of cortical regions both at the cortico–cortical and cortico–STN level. Still, significant connectivity was selectively preserved in a spectrally specific manner ON medication both at the cortico–cortical (sensorimotor–frontoparietal network) and the cortico–STN levels. The most surprising aspect of this state was the emergence of bilateral STN–STN coherence ON medication across all frequency modes.

## Dopamine selectively modifies delta/theta oscillations within the STN–STN state

In this STN–STN state, dopaminergic intervention had only a limited effect on STN–STN connectivity. OFF medication, STN–STN coherence was present across all three frequency modes (p<0.05, *Figure 4A*), while ON medication, significant STN–STN coherence emerged only in the alpha and beta band (p<0.05, *Figure 4C* alpha and beta). ON medication, STN–STN delta/theta connectivity strength was not significantly different from the mean noise level (p<0.05, *Figure 4C* delta).

OFF compared to ON medication, coherence was reduced across the entire cortex both at the inter-cortical and the STN–cortex level across all frequency modes. The most affected areas were similar to the ones in the Ctx–STN state, in other words, the sensorimotor, frontal, and temporal regions. Their coherence with the STN was also significantly reduced, ON compared to OFF medication (STN–sensorimotor, p<0.01 delta/theta, beta; p<0.05 alpha; STN–temporal, p<0.01 delta/theta, alpha, beta; and STN–frontal, p<0.01 delta/theta, alpha and beta; *Figure 4B*).

In summary, STN–STN connectivity was not significantly altered OFF to ON medication. At the same time, coherence decreased from OFF to ON medication at both the cortico–cortical and the cortico–STN level. Therefore, only significant STN–STN connectivity existed both OFF and ON medication, while cortico–STN or cortico–cortical connectivity changes remained at the mean noise level.

## States with a generic coherence decrease have longer lifetimes

Using the temporal properties of the identified networks, we investigated whether states showing a shift towards physiological connectivity patterns lasted longer ON medication. A state that is physiological should exhibit increased lifetime and/or should occur more often ON medication. An example of the state time courses is shown in *Figure 5*.

*Figure 6A-C* shows the temporal properties for the three states for both the OFF and ON medication conditions. Two-way repeated measures ANOVA on the temporal properties of the HMM

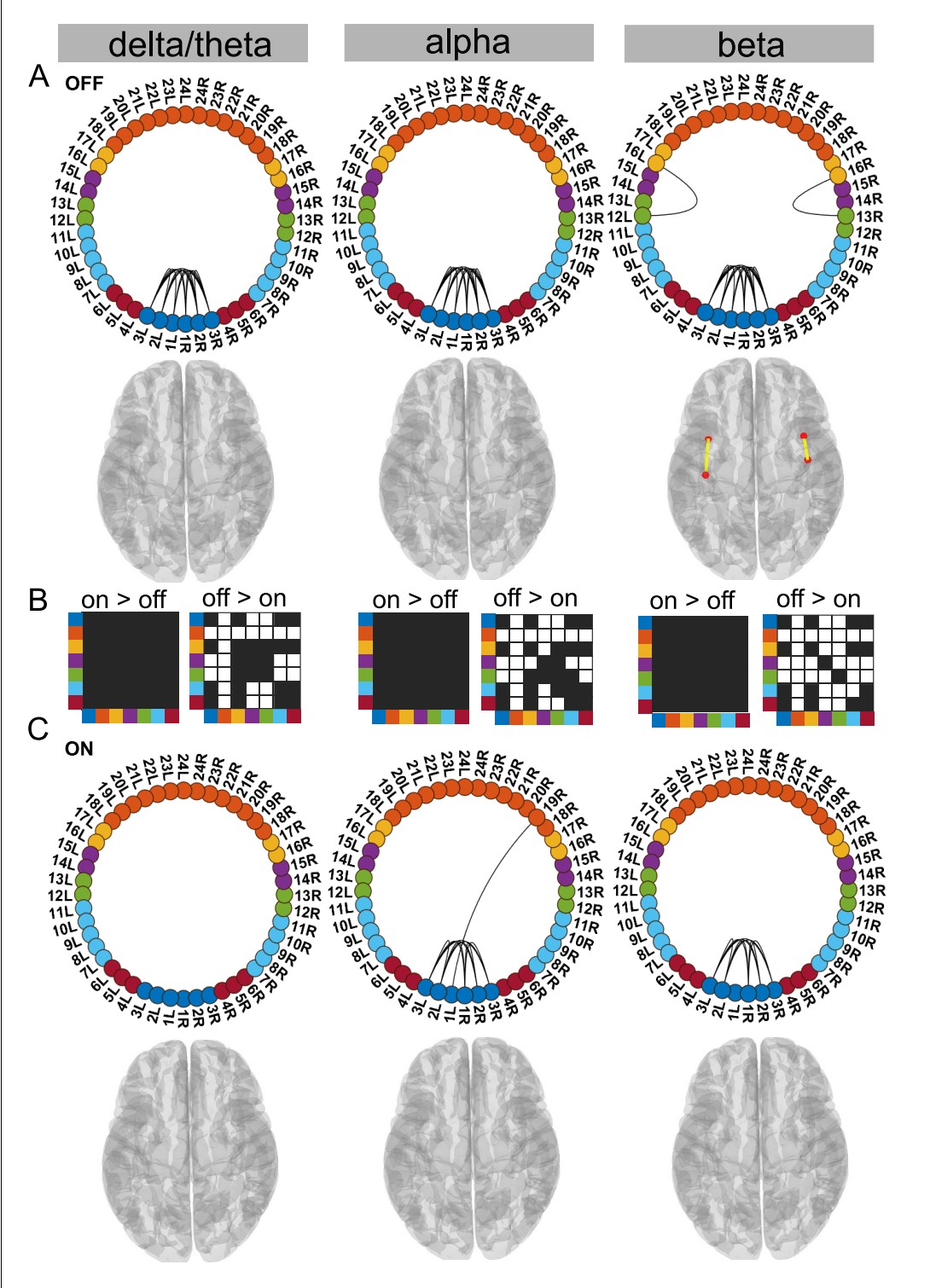

**Figure 4.** STN–STN state. For the general description, see the note to *Figure 2*. The STN–STN state was characterised by preservation of STN–STN coherence in the alpha and beta band OFF versus ON medication. STN–STN theta/delta coherence was no longer significant ON medication. Source data are provided as *Figure 4—source data 1–3*.

The online version of this article includes the following source data for figure 4:

**Source data 1.** Source data of *Figure 4a*.
**Source data 2.** Source data of *Figure 4b*.
**Source data 3.** Source data of *Figure 4c*.

**Table 1.** Regions of the Mindboggle atlas used.

STN, subthalamic nucleus; Vis, visual; Par, parietal; Smtr, sensory motor; Tmp, temporal; Mpf, medial prefrontal; Frnt, frontal; Ctx, cortex. The colour code is for the ring figures presented as part of the results.

| STN | 1 | Contact one right | Smtr-Ctx | 12 | Postcentral |
|---|---|---|---|---|---|
| | 2 | Contact two right | | 13 | Precentral |
| | 3 | Contact three right | Tmp-Ctx | 14 | Middle temporal |
| | 1 | Contact four left | | 15 | Superior temporal |
| | 2 | Contact five left | Mpf-Ctx | 16 | Caudal middle frontal |
| | 3 | Contact six left | | 17 | Medial orbitofrontal |
| Vis-Ctx | 4 | Cuneus | Frnt-Ctx | 18 | Insula |
| | 5 | Lateral occipital | | 19 | Lateral orbitofrontal |
| | 6 | Lingual | | 20 | Pars opercularis |
| Par-Ctx | 7 | Inferior parietal | | 21 | Pars orbitalis |
| | 8 | Para central | | 22 | Pars triangularis |
| | 9 | Precuneus | | 23 | Rostral middlefrontal |
| | 10 | Superior parietal | | 24 | Superior frontal |
| | 11 | Supramarginal | | | |

states revealed an effect of HMM states on the fractional occupancy (FO) ($F_{(2,96)}=10.49$, $p<0.01$), interval of visits ($F_{(2,221513)}=9783.13$, $p<0.01$), and lifetime ($F_{(2,214818)}=50.36$, $p<0.01$). There was no effect of medication (L-DOPA) on FO ($F_{(1,96)}=2.00$, $p=0.16$) and lifetime ($F_{(1,214818)}=0.15$, $p=0.7026$). Medication had a significant effect on the interval of visits ($F_{(1,221513)}=4202.96$, $p<0.01$). Finally, we found an interaction between the HMM states and medication on the interval of visits ($F_{(2,221513)}=1949.98$, $p<0.01$) and lifetime ($F_{(2,214818)}=172.25$, $p<0.01$). But there was no interaction between HMM states and medication on FO ($F_{(2,96)}=0.54$, $p=0.5855$).

We performed post hoc testing on the ANOVA results. OFF medication, the STN–STN state was the one with the longest lifetime (STN–STN >Ctx Ctx, $p<0.01$; STN–STN >Ctx-STN, $p<0.01$). The Ctx–STN state OFF medication had the shortest lifetime among all three states (Ctx–STN <Ctx-Ctx, $p<0.01$; Ctx–STN <STN-STN, $p<0.01$) and the shortest interval between visits (interval of visit Ctx–STN <Ctx-Ctx, $p<0.01$; Ctx–STN <STN-STN, $p<0.01$). The largest interval between visits was for the Ctx–Ctx state OFF medication (Ctx–Ctx >STN-STN, $p<0.01$; Ctx–Ctx >Ctx-STN, $p<0.01$). The FO for the STN–STN and Ctx–STN states was similar, but significantly higher than for the Ctx–Ctx state (STN–STN >Ctx-Ctx, $p<0.01$; STN–STN $\approx$ Ctx–STN, $p=0.82$; Ctx–STN >Ctx-Ctx, $p<0.01$). ON medication, the comparison between temporal properties of all three states retained the same significance levels as OFF medication, except for the lifetime of the Ctx–STN state, which was no longer significantly different from that of the Ctx–Ctx state ($p=0.98$). Within each medication condition, the states retained their temporal characteristics relative to each other.

Across medication conditions, significant changes were present in the temporal properties of the states. The lifetimes for both the STN–STN and Ctx–STN state were significantly increased by medication (ON >OFF: STN–STN, $p<0.01$; Ctx–STN, $p\leq0.01$) but the lifetime for the Ctx–Ctx state was not significantly influenced by medication. The Ctx–Ctx state was visited even less often ON medication (interval: ON >OFF Ctx–Ctx, $p<0.01$). The interval between visits remained unchanged for the STN–STN and Ctx–STN states. The FO for all three states was not significantly changed from OFF to ON medication. In summary, the cortico–cortical state was visited least often compared to the other two states both OFF and ON medication. The cortico–STN and STN–STN states showing physiologically relevant spectral connectivity lasted significantly longer ON medication.

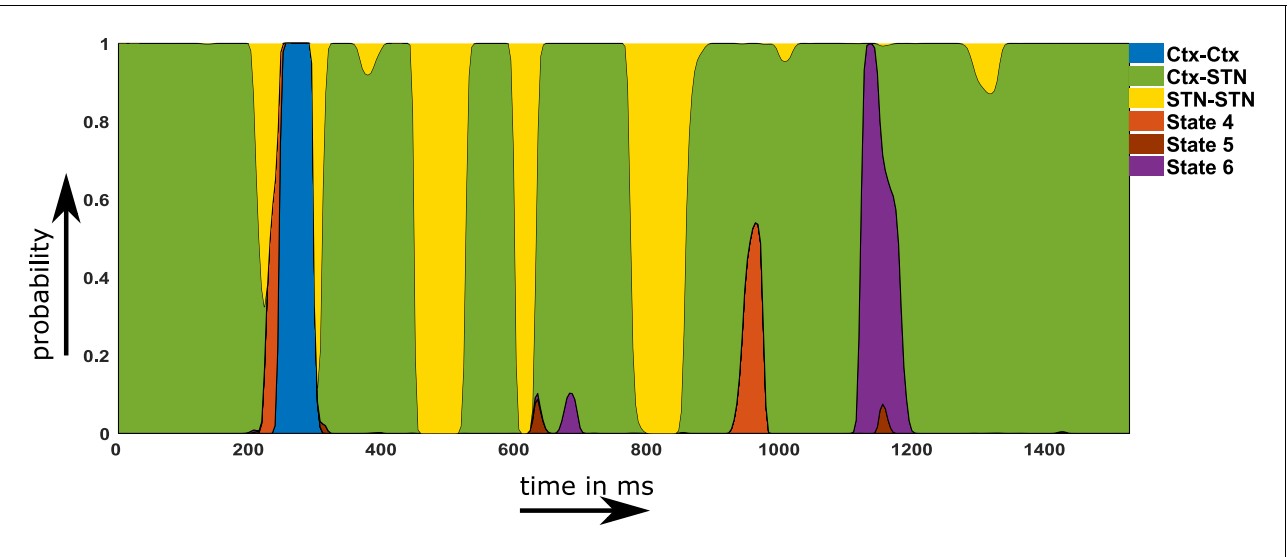

**Figure 5.** Example of a probability time course for the six hidden Markov model (HMM) states OFF medication. Note that within the main text of the paper, we are only discussing the first three states. The connectivity maps of states 4–6 are provided in *Figure 2—figure supplement 1*. Source data are provided as *Figure 5—source data 1–2*.
The online version of this article includes the following source data for figure 5:

**Source data 1.** Probability time course first half in relation to *Figure 5*.
**Source data 2.** Probability time course second half in relation to *Figure 5*.

## Discussion

In this study, we parsed simultaneously recorded MEG–STN LFP signals into discrete time-resolved states to reveal distinct spectral communication patterns. We identified three states exhibiting distinct coherence patterns ON and OFF medication: a cortico–cortical, a cortico–STN, and a STN–STN state. Our results indicate a tendency of neural activity to engage in connectivity patterns in which coherence decreases under the effect of dopaminergic medication and which maintain selective cortico–STN connectivity (Ctx–STN and STN–STN states). Only within the Ctx–Ctx state did coherence increase under dopaminergic medication. These results are in line with the multiple effects of dopaminergic medication reported in resting and task-based PD studies (*Jubault et al., 2009*; *West et al., 2016*; *Tinkhauser et al., 2017*).

The differential effect of dopamine allowed us to delineate pathological and physiological spectral connectivity. The Ctx–Ctx state provided electrophysiological evidence in the delta/theta band for the overdose effect of dopaminergic medication in PD. Prior to the electrophysiological evidence in our study, there was only evidence through task-based or functional magnetic resonance imaging (fMRI) studies (*Cools et al., 2002*; *Ray and Strafella, 2010*; *MacDonald et al., 2011*). The Ctx–STN state revealed that simultaneous cortico–cortical and STN–STN interactions emerge ON medication, with spectrally and spatially specific cortico–STN interactions. In addition, ON medication, a frontoparietal motor network was present, indicating a shift from STN-mediated motor connectivity to a cortical one. These findings have not been reported in previous studies. The STN–STN state exhibited the limited ability of dopaminergic medication to modify local STN–STN delta oscillations. Our analysis also revealed significant changes in the temporal properties of the connectivity profiles, including lifetime and FO, under the effect of dopaminergic medication. This insight might in the future prove important for modifying medication as well as DBS-based strategies for therapeutic purposes.

### Increased tonic dopamine causes excessive frontal cortical activity

The Ctx–Ctx state showed significant coherent connectivity between the orbitofrontal cortical regions in the delta/theta band ON medication. According to the dopamine overdose

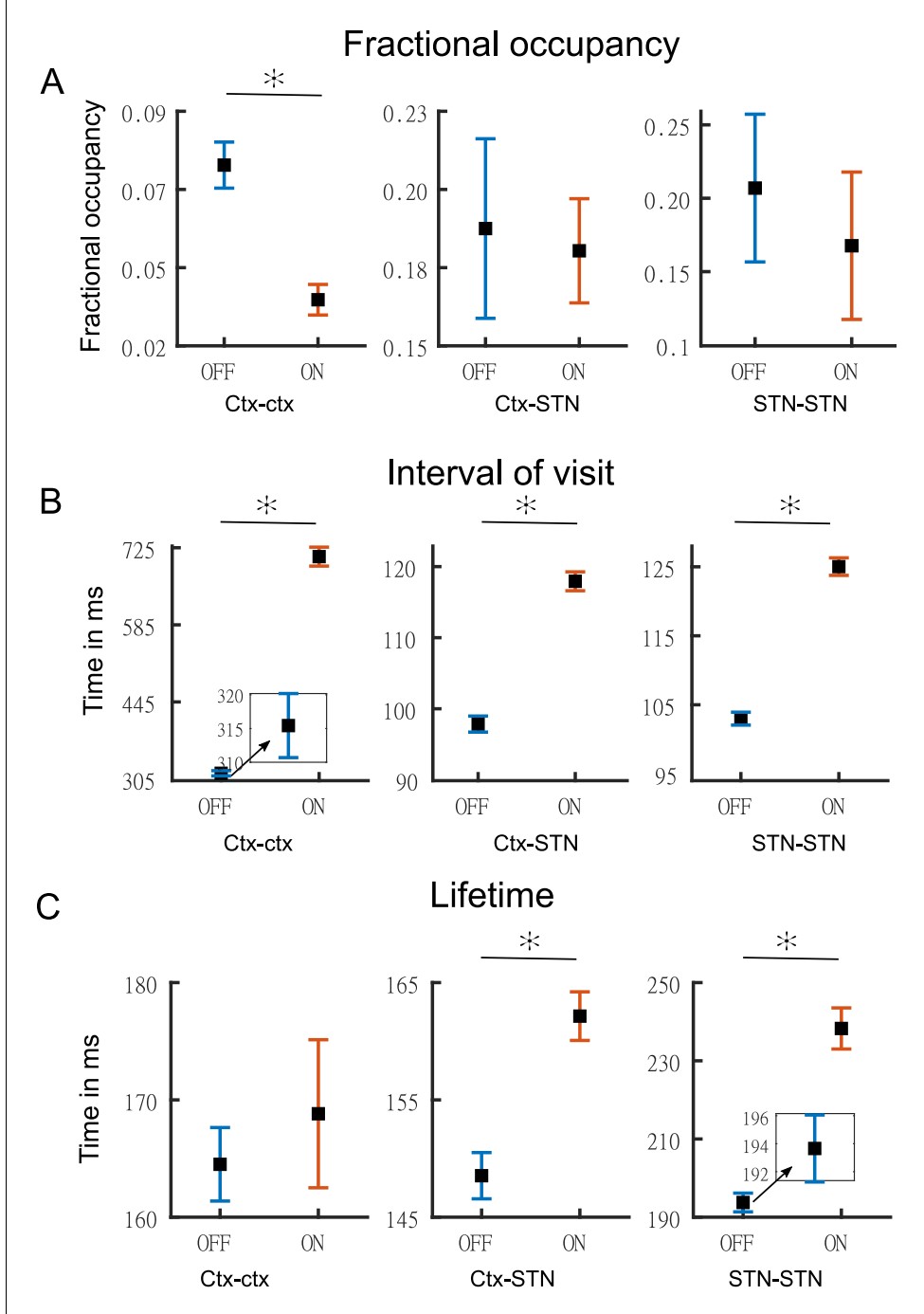

**Figure 6.** Temporal properties of states. Panel **A** shows the fractional occupancy for the three states for the cortico–cortical (Ctx–Ctx), cortico–STN (Ctx–STN), and the STN–STN (STN–STN). Each point represents the mean for a state and the error bar represents standard error. Orange denotes ON medication data and blue OFF medication data. Panel **B** shows the mean interval of visits (in milliseconds) of the three states ON and OFF medication. Panel **C** shows the lifetime (in milliseconds) for the three states. Figure insets are used for clarity in case error bars are not clearly visible. The y-axis of each figure inset has the same units as the main figure. Source data are provided as *Figure 6—source data 1–6*.

The online version of this article includes the following source data for figure 6:

**Source data 1.** Source data of *Figure 6a* OFF medication.
**Source data 2.** Source data of *Figure 6a* ON medication.
**Source data 3.** Source data of *Figure 6b* OFF medication.
**Source data 4.** Source data of *Figure 6b* ON medication.

**Source data 5.** Source data of *Figure 6c* OFF medication.
**Source data 6.** Source data of *Figure 6c* ON medication.

hypothesis in PD (*Cools, 2001*; *Kelly et al., 2009*; *MacDonald and Monchi, 2011*; *Vaillancourt et al., 2013*), the commonly used doses of dopaminergic medication to mitigate the motor symptoms cause the ventral frontostriatal cortical circuits to experience an excessive increase in tonic dopamine levels. This medication-induced increase is due to excessive compensation of dopamine in the ventral striatal circuitry, which experiences a lower loss of dopamine than its dorsal counterpart. The reason is that in PD dopaminergic neurons in the substantia nigra are primarily lost and therefore the dopamine depletion within the dorsal circuitry is higher than within the ventral one (*Kelly et al., 2009*; *MacDonald and Monchi, 2011*). Frontal regions involved with the ventral striatal circuitry include the orbitofrontal cortex, anterior cingulate, and the inferior temporal cortex (*Cools, 2006*; *MacDonald and Monchi, 2011*). Increased frontal cortex connectivity potentially explains the cognitive deficits observed in PD (*Shohamy et al., 2005*; *George et al., 2013*). Our detected emergence of frontal cortico–cortical coherence (between orbitofrontal and medial orbitofrontal regions) specifically in the delta/theta band could explain the cognitive deficits observed in PD due to dopaminergic medication, given the role of frontal delta/theta oscillations in cognition (*Harmony, 2013*; *Zavala et al., 2014*).

A comparison of temporal properties of the Ctx–Ctx state OFF versus ON medication revealed that the interval between visits was significantly increased ON medication, while the FO of this state was significantly reduced. In fact, the FO of the Ctx–Ctx state was the lowest among the three states. The temporal results indicate that the Ctx–Ctx state is least visited. Neural activity ON medication is not likely to visit this state, but whenever it does, its visits are of the same duration as OFF medication. Hence, the Ctx–Ctx state's presence could explain the cognitive side effects observed ON medication in PD.

## Selective spectral connectivity remains preserved with increased dopamine levels

An interesting feature of the Ctx–STN state was the emergence of local STN–STN coherence in all three frequency modes. Bilateral STN–STN coherence in the alpha and beta band did not change in the Ctx–STN state ON versus OFF medication (InterMed analysis). However, STN–STN coherence was significantly higher than the mean level ON medication (IntraMed analysis). Since synchrony limits information transfer (*Cruz et al., 2009*; *Cagnan et al., 2015*; *Holt et al., 2019*), the high coherence within the STN ON medication could prevent communication with the cortex. A different explanation would be that a loss of cortical afferents leads to increased local STN coherence. The causal nature of the cortico–basal ganglia interaction is an endeavour for future research.

Previous studies have reported STN–sensorimotor (*Hirschmann et al., 2011*; *Litvak et al., 2011*), STN–parietal, and STN–frontal (*Litvak et al., 2011*) coherence in the beta band OFF medication. Consistent with previous studies STN–sensorimotor, STN–parietal (inferior parietal), STN–frontal (insular cortex, pars orbitalis, pars opercularis, and lateral orbitofrontal), and STN–medial prefrontal (medial orbitofrontal) coherence emerged in the Ctx–STN state. In contrast, ON medication sensorimotor regions were coherent with parietal (para central) and frontal (superior frontal)/medial prefrontal (caudal middle frontal) regions in the beta frequency range. Previous research has not reported the emergence of such a coherent frontoparietal motor network ON medication. But consistent with previous research (*Hirschmann et al., 2013*), sensorimotor–STN coherence was reduced ON compared to OFF medication.

In addition, critical processing regarding sensorimotor decision-making involves frontoparietal regions (*Gertz and Fiehler, 2015*; *Siegel et al., 2015*; *Gallivan et al., 2018*; *Martínez-Vázquez and Gail, 2018*). Hence, the emergence of frontoparietal connectivity with motor regions points towards the physiological relevance of the Ctx–STN state. Moreover, neural activity ON medication remained longer in the Ctx–STN state as the lifetime of this state significantly increased compared to OFF medication. The finding is in line with our hypothesis that a state showing physiologically relevant spectral connectivity lasts longer ON medication.

## Tonic dopamine has a limited effect on local STN–STN interactions

In the Ctx–STN state, STN–STN coherence accompanied network changes affecting cortico–STN communication ON medication, thereby likely having a functional role. In contrast, in the STN–STN state, STN–STN coherence emerged without the presence of any significant cortico–STN coherence either OFF or ON medication. This may indicate that the observed STN–STN activity in the STN–STN state emerged due to local basal ganglia circuitry. No changes were observed in the alpha and beta band in the STN–STN state ON versus OFF medication, which may indicate the inability of tonic dopamine to modify basal ganglia circuit activity. These results provide more evidence that the changes in STN–STN coherence observed in previous studies (*Little et al., 2013*; *Oswal et al., 2013*; *Shimamoto et al., 2013*) reflect cortical interaction affecting STN activity. Future studies should analyse changes occurring within the STN. To the best of our knowledge, we are the first to uncover modulation of STN–STN delta/theta oscillations by dopaminergic medication. Studies have shown that local basal ganglia delta oscillations, which do not require input from the motor cortex, are robust biomarkers of dopamine depletion (*Whalen et al., 2020*). Hence, selective elimination of delta/theta oscillations under dopaminergic medication in the STN–STN state points towards restoration of physiologically relevant network activity.

## Limitations of the study

In the present study, we employed a data-driven approach based on an HMM. In order to find the appropriate model, we had to specify the number of states a priori. We selected the number of HMM states based on a compromise between spectral quality of results and their redundancy. The number of states could also be determined by selecting the one with the highest negative free energy. However, model selection based on free energy often does not yield concrete results (*Baker et al., 2014*). Another limitation is the use of multivariate Gaussian distributions to characterise the state covariance matrices. Although it improves the tractability of the HMM estimation process, it is by construction unable to capture higher-order statistics of the data beyond the first two moments. For example, burst activity might also be a relevant property of brain networks (*Florin et al., 2015*). Lastly, we would like to note that the HMM was used as a data-driven, descriptive approach without explicitly assuming any a priori relationship with pathological or physiological states. The relation between biology and the HMM states, thus, purely emerged from the data; that is, is empirical. What we claim in this work is simply that the features captured by the HMM hold some relation with the physiology even though the estimation of the HMM was completely unsupervised (i.e., blind to the studied conditions).

Besides these limitations inherent in the analysis approach, there are also some related to the experimental design. As this is a study containing invasive LFP recordings, we can never have a healthy control group. In addition, we only recorded four female patients because during the study period fewer female patients underwent a DBS surgery at our centre. To the best of our knowledge, there is no previous literature reporting a sex difference in MEG markers or the prescribed dopaminergic medication (*Umeh et al., 2014*). The medication led to a marked motor improvement in these patients based on the UPDRS, but the patients still have impairments. Both motor impairment and motor improvement can cause movement during the resting state in PD. While such movement is a deviation from a resting state in healthy subjects, such movements are part of the disease and occur unwillingly. Therefore, such movements can arguably be considered part of the resting state of PD. None of the patients in our cohort experienced hyperkinesia during the recording. All patients except for two were of the akinetic-rigid subtype. We verified that tremor movement is not driving our results. Recalculating the HMM states without these two subjects, even though it slightly changed some particular aspects of the HMM solution, did not materially affect the conclusions. A further potential influencing factor might be the disease duration and the amount of dopamine patients are receiving. Both factors were not significantly related to the temporal properties of the states.

To differentiate pathological and physiological network activity, we had to rely on the temporal properties of the networks. A further limitation was that all our recordings were made under resting conditions, preventing us from discerning the functional role of oscillations within the discovered networks. We opted for the current design because resting-state data allows the study of networks independent of a task and because using a specific task bears the risk that the patients are not able

to properly perform it. Nevertheless, future studies should analyse the behaviour of specific networks using tasks to probe them.

Lastly, we recorded LFPs from within the STN – an established recording procedure during the implantation of DBS electrodes in various neurological and psychiatric diseases. Although for Parkinson patients results on beta and tremor activity within the STN have been reproduced by different groups (*Reck et al., 2010*; *Litvak et al., 2011*; *Florin et al., 2013*; *Hirschmann et al., 2013*; *Neumann et al., 2016*), it is still not fully clear whether these LFP signals are contaminated by volume-conducted cortical activity. However, while volume conduction seems to be a larger problem in rodents even after re-referencing the LFP signal (*Lalla et al., 2017*), the same was not found in humans (*Marmor et al., 2017*). Moreover, we used directional contacts, which have a smaller surface area than the classical ring contacts. Based on the available literature, our sampling rate was high enough to resolve oscillatory activity in the STN (*Telkes et al., 2020*; *Nguyen et al., 2020*).

## Conclusion

Using a data-driven machine learning approach, we identified three distinct networks (states) that captured differential effects of dopaminergic medication on spectral connectivity in PD. Our findings uncovered a Ctx–Ctx state that captured the potentially adverse effects of increased dopamine levels due to dopaminergic medication. Furthermore, a Ctx–STN state was identified that maintained spatio-spectrally selective cortico–STN connectivity ON medication. We also found an STN–STN coherent state, pointing towards the limited effect of dopaminergic medication to modify local basal ganglia activity. Our findings bring forth a dynamical systems perspective for differentiating pathological versus physiologically relevant spectral connectivity in PD. Furthermore, we were able to uncover differential changes induced by altered levels of a neuromodulator such as dopamine in a completely data-driven manner without providing detailed information about large-scale dopaminergic networks to the HMM. This shows another advantage of our dynamical systems-level approach. Furthermore, our whole-brain STN approach provides novel electrophysiological evidence of distributed changes due to dopaminergic medication in brain connectivity, extending previous pairwise connectivity results reported in PD.

# Materials and methods

## Subjects

In total, 17 (4 female) right-handed PD patients (age: 55.2 ± 9.3 years) undergoing surgery for therapeutic STN DBS were recruited for this study. Patients had been selected for DBS treatment according to the guidelines of the German Society for Neurology. The experimental procedure was explained to all participants and they gave written consent. The study was approved by the local ethics committee (study number 5608R) and conducted in accordance with the Declaration of Helsinki. Bilateral DBS electrodes were implanted in the dorsal part of the STN at the Department of Functional Neurosurgery and Stereotaxy in Düsseldorf. The implanted DBS electrodes used were the St. Jude Medical directional lead 6172 (Abbott Laboratories, Lake Bluff, IL) and in one case the Boston Scientific Vercise segmented lead (Boston Scientific Corporation, Marlborough, MA). These electrodes have four contact heights and the two middle heights are segmented into three equally spaced contacts.

The DBS leads were externalised and we measured the patients after 1–3 days. To simultaneously acquire MEG and LFP signals, we connected the externalised leads to an EEG amplifier integrated with the MEG system. We used a whole-head MEG system with 306 channels (Elekta Vectorview, Elekta Neuromag, Finland) housed within a magnetically shielded chamber. All patients were requested to sit still and awake during data acquisition. To ensure that patients did not fall asleep, we tracked patients' pupil diameter with an eye tracker. To remove eye blink and cardiac artefacts, electrooculography and electrocardiography were recorded along with the LFP and MEG signals. In order to co-register the MEG recording with the individual MRI, four head position indicator coils were placed on the patient's head. Their position as well as additional head points were digitised using the Polhemus Isotrack system (Polhemus, Colchester, CT). The data were recorded with a sampling rate of 2400 Hz and a low-pass filter of 800 Hz was applied. An electrode was placed at the mastoid and all LFP signals were referenced to it.

For the clinical OFF medication state, oral PD medication was withdrawn overnight for at least 12 hr. If a patient had an apomorphine pump, this pump was stopped at least 1 hr before the measurement. First, we recorded resting-state activity in the medication OFF condition. The patients were then given their morning dose of L-DOPA in the form of fast-acting levodopa. Data were acquired in three runs of 10 min, for a total of 30 min for each medication condition. We started the ON medication measurement at least half an hour after the administration of the dose and after clinical improvement was seen. The same procedure as for the OFF medication state was followed for the ON medication measurement.

## Pre-processing

All data processing and analyses were performed using Matlab (version R 2016b; Math Works, Natick, MA). Custom-written Matlab scripts (https://github.com/saltwater-tensor/HMM_pipeline (copy archived at swh:1:rev:277a6a0ff21ff6885815c934255f953a97e16e98)); *Sharma et al., 2021a Sharma et al., 2021b* and the Brainstorm toolbox (http://neuroimage.usc.edu/brainstorm/Introduction) were used (*Tadel et al., 2011*). To ensure artefact-free data, two people independently inspected the data visually, cleaned artefacts, and compared the cleaning output. The final cleaned data included changes agreed upon by both the people involved in cleaning. The Neuromag system provides signal-space projection (SSP) vectors for the cleaning of external artefacts from the MEG channels, which were applied. The line noise was removed from all channels with a notch filter at 50, 100, 150, . . ., 550, and 600 Hz with a 3 dB bandwidth of 1 Hz. The LFP recordings from the DBS electrode were re-referenced against the mean of all LFP channels. Very noisy and flat MEG/LFP channels were excluded from further analysis. Time segments containing artefacts were removed from the time series. However, if artefacts regularly occurred only in one single channel, this whole channel was removed instead. Frequently arising artefacts following the same basic pattern, such as eye blinks or cardiac artefacts, were removed via SSP. All data were high-pass filtered with 1 Hz to remove movement-related low-frequency artefacts. Finally, the data were down-sampled to 1000 Hz.

Source estimation was performed on these recordings at an individual level using each individual's anatomy. Therefore, using Freesurfer (https://surfer.nmr.mgh.harvard.edu/, v.5.3.0), the individual cortical surfaces were extracted from the individual T1-weighted MRI scans (3T scanner and 1 $mm^3$ voxel size). We used the overlapping spheres method with 306 spheres for the forward model. As the inverse model, we used a linearly constrained minimum variance (LCMV) beamformer. The data covariance matrix for the LCMV beamformer was computed directly from each 10 min recording. The data covariance was regularised using the median eigenvalue of the data covariance matrix. The noise covariance was obtained from an empty room recording on the same day as the actual measurement.

For each subject, the invasive entry point of the STN was identified based on intraoperative microelectrode recordings (*Gross et al., 2006*; *Moran et al., 2006*). Subsequently, the first recording height after the entry into the STN was selected to obtain the three directional LFP recordings from the respective hemisphere. In addition, we visualised the location of all electrodes using lead-DBS (*Horn et al., 2019*). All electrodes were properly placed within the STN – except for one (see *Figure 7*). To exclude that our results were driven by outlier, we reanalysed our data without this patient. No qualitative change in the overall connectivity pattern was observed.

The source-reconstructed MEG data were projected to the default cortical anatomy (MNI 152 with 15,002 vertices) and then down-sampled temporally to 250 Hz for each medication condition for every subject. We used the Mindboggle atlas to spatially reduce the data dimensions. For each of the 42 cortical regions in the atlas, a multidimensional time series consisting of the vertices within that anatomical region was extracted. To reduce the multivariate times series for each region to a single one, we employed the first principal component explaining the highest variance share in each region. The first principal component row vectors from all 42 anatomical regions were stacked into a MEG cortical time series matrix. To correct for volume conduction in the signal, symmetric orthogonalisation (*Colclough et al., 2015*) was applied to each subject's resulting MEG cortical time series matrix. The row vectors of this orthogonalised matrix and the six LFPs (three each for left and right STN) were z-scored. Subsequently, they were stacked into one multidimensional time series (N by T) matrix. Here, N = 48 is the total number of nodes/regions (42 regions from the cortex and 6 LFP electrode contacts) and T denotes the length of the time dimension. This 48 by T data matrix

obtained from each subject was concatenated along the temporal dimension across all subjects for each specific medication condition. Finally, to resolve sign ambiguity inherent in source-reconstructed MEG data as well as resolve polarity of LFP channels across subjects, a sign-flip correction (*Vidaurre et al., 2016*) procedure was applied to this final 48 by (T by number of subjects) dataset within a medication condition. The pre-processing steps were performed for OFF and ON medication separately.

## HMM analysis

The HMM is a data-driven probabilistic algorithm which finds recurrent network patterns in multivariate time series (*Vidaurre et al., 2016*; *Vidaurre et al., 2018a*). Each network pattern is referred to as a 'state' in the HMM framework, such that these networks can activate or deactivate at various points in time. Here onwards, 'state' or 'network' is used interchangeably. We used a specific variety of the HMM, the TDE-HMM, where whole-brain networks are defined in terms of both spectral power and phase coupling (*Vidaurre et al., 2018b*). Hence, for every time point, the HMM algorithm provided the probability that a network is active. Here onwards, a contiguous block of time for which the probability of a particular network being active remained higher than all the other networks is referred to as a 'state visit'. Hence, the HMM produced temporally resolved spatial networks for the underlying time series. In our approach, we also performed spectral analyses of these state visits, leading to a complete spatio-spectral connectivity profile across the cortex and the STN. By applying the HMM analysis to the combined MEG–LFP dataset, we were able to temporally, spatially, and spectrally separate cortico–cortical, cortico–STN, and STN–STN networks.

## Estimation of the HMM

Since we were interested in recovering phase-related networks, the TDE-HMM was fit directly on the time series obtained after pre-processing steps described previously, as opposed to its power envelope. This preserved the cross-covariance within and across the underlying raw time series of the cortical regions and the STN. The model estimation finds recurrent patterns of covariance between regions (42 cortical regions and 6 STN contacts) and segregates them into 'states' or 'networks'.

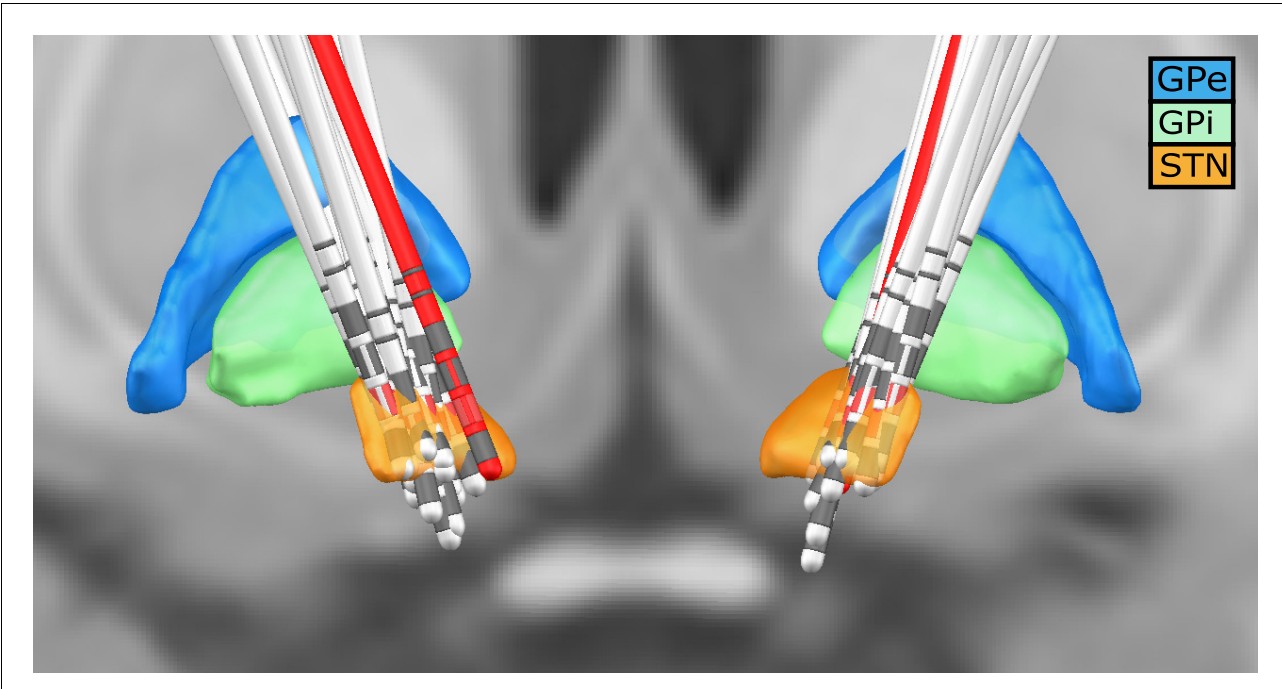

**Figure 7.** Deep brain stimulation (DBS) electrode location for all subjects. Lead-DBS reconstruction with all subjects. The red leads are the ones of a subject with one of the outside the STN. The red directional contacts are the ones from which the data was used for analysis.

Based on these covariance patterns, for each state, the power spectra of each cortical region and the coherence amongst regions can be extracted.

We opted for six different states as a reasonable trade-off between the spectral quality of the results and their redundancy. The HMM-MAR toolbox (*Vidaurre et al., 2016*) was used for fitting the TDE-HMM. We employed the TDE version of the HMM where the embedding took place in a 60 ms window (i.e., a 15 time point window for a sampling frequency of 250 Hz). Since time embedding would increase the number of rows of the data from 48 to 48 times the window length (also referred to as number of lags), an additional PCA (principal component analysis) (reduction across 48 by number of lags) step was performed after time embedding. The number of components retained was 96 (48 × 2). This approach follows *Vidaurre et al., 2018b*. To characterise each state, a full covariance matrix with an inverse Wishart prior was used. The diagonal of the prior for the transition probability matrix was set as 10. To ensure that the mean of the time series did not take part in driving the states, the 'zero mean' option in HMM toolbox was set to 1. To speed up the process of fitting, we used the stochastic version of variational inference for the HMM. In order to start the optimisation process, the 'HMM-MAR'-type initialisation was used (for details, see *Vidaurre et al., 2016*). The HMM was fit separately OFF and ON medication.

## Statistical analysis of the states

After the six states were obtained for HMM OFF and HMM ON medication, these states were statistically compared within each medication condition as well as between medication conditions. In addition, the temporal properties of these states were compared.

## Intra-medication analysis

We investigated the spectral connectivity patterns across the different states within a medication condition (intra-medication or IntraMed). The objective was to uncover significant coherent connectivity standing out from the background within each frequency band (delta/theta [1–8 Hz], alpha [8–12 Hz], and beta [13–30 Hz]) in the respective states. The HMM output included the state time courses (i.e., when the states activated) for the entire concatenated data time series. The state time courses allowed the extraction of state- and subject-specific data for further state- and subject-level analysis. For each HMM state, we filtered the state-specific data for all the subjects between 1 and 45 Hz. (For state-wise data extraction, please refer the HMM toolbox wiki [https://github.com/OHBA-analysis/HMM-MAR/wiki/User-Guide].) Then we calculated the Fourier transform of the data using a multitaper approach to extract the frequency components from the short segments of each state visit. (See *Vidaurre et al., 2018b* for discussion on multitaper for short time data segments.) Seven Slepian tapers with a time–bandwidth product of 4 were used, resulting in a frequency resolution of 0.5 Hz and therefore binned frequency domain values. Subsequently, we calculated the coherence and power spectral density of this binned (frequency bins obtained during the multitaper step) data for every subject and every state. The coherence and the power spectral density obtained were three-dimensional matrices of size f (number of frequency bins) by N (42 cortical locations + 6 STN contacts) by N.

Based on the coherence matrices, we performed a frequency band-specific analysis. Canonical definitions of frequency bands assign equal weight to each frequency bin within a band for every subject. This might not be suitable when considering analyses of brain signals across a large dataset. For example, the beta peak varies between individual subjects. Assigning the same weight to each bin in the beta range might reduce the beta effect at the group level. To allow for inter-subject variability in each frequency bin's contribution to a frequency band, we determined the frequency modes in a data-driven manner (*Vidaurre et al., 2018b*). Because we focused on interactions that are important to establish the STN–cortex communication, the identification of the relevant frequency modes was restricted to the cross-coherence between the STN–LFPs and cortical signals; in other words, the block matrix consisting of rows 1–6 (STN) and columns 7–48 (cortex). For each subject, this extracted submatrix was then vectorised across columns. This gave us a (number of frequency bins by 252 [6 STN locations by 42 cortical locations]) matrix for each state. For every subject, this matrix was concatenated along the spatial dimension across all states producing a (number of frequency bins by [252 by 6 (number of states)]) matrix. We called this the subject-level coherence matrix. We averaged these matrices across all subjects along the spectral dimension (number

of frequency bins) to yield a (number of frequency bins by [252 by 6]) group-level coherence matrix. We factorised the group-level coherence matrix into four frequency modes using a non-negative matrix factorisation (NNMF) (*Lee and Seung, 2001*). Each of the resulting four frequency modes obtained was of size (one by number of frequency bins). The values of frequency modes are the actual NNMF weights obtained from the NNMF estimation (which, just like a regression coefficient, are unit-less, because coherence is unit-less). Three of them resembled the canonical delta/theta (delta and theta frequencies were combined into one band), alpha, and beta bands whereas the last one represented noise. Since NNMF does not guarantee a unique solution, we performed multiple instances of the factorisation. In practice we could obtain frequency modes, which showed correspondence to the classical frequency bands, within four iterations of the algorithm. At each instance, we visualised the output to ensure frequency specificity of the frequency modes. The stability of the output was ensured by using 'robust NNMF', which is a variant of the NNMF algorithm (*Vidaurre et al., 2018b*). While these frequency modes were derived in fact from coherence measures (as detailed in *Vidaurre et al., 2018a*), they can be applied to power measures or any other frequency-specific measure. We then computed the inner product between the subject- and group-level coherence matrix and the frequency modes obtained above. We called these the subject-level and group-level projection results, respectively.

To separate background noise from the strongest coherent connections, a Gaussian mixture model (GMM) approach was used (*Vidaurre et al., 2018b*). For the group-level projection results, we normalised the activity in each state for each spectral band by subtracting the mean coherence within each frequency mode across all states. As a prior for the mixture model, we used two single-dimensional Gaussian distributions with unit variance: one mixture component to capture noise and the other to capture the significant connections. This GMM with two mixtures was applied to the coherence values (absolute value) of each state. Connections were considered significant if their p-value after correction for multiple comparisons was smaller than 0.05.

## Inter-medication analysis

To test for differences in coherence across medication conditions (inter-medication or InterMed), the first step was to objectively establish a comparison between the states found in the two HMMs fit separately for each condition. There is no a priori reason for the states detected in each condition to resemble each other. To find OFF and ON medication states that may resemble each other, we calculated the Riemannian distance (*Förstner and Moonen, 2003*) between the state covariance matrices of the OFF and ON HMM. This yielded an OFF states by ON states ($6 \times 6$) distance matrix. Subsequently, finding the appropriately matched OFF and ON states reduced to a standard linear assignment problem. We found an ON state counterpart to each OFF state by minimising the total sum of distances using the Munkres linear assignment algorithm (*Vidaurre et al., 2018a*). This approach yielded a one-to-one pairing of OFF and ON medication states, and all further analysis was conducted on these pairs. For ease of reading, we gave each pair its own label. For example, when we refer to a 'Ctx–STN' state in the following sections, then such a state was discovered OFF medication and its corresponding state ON medication is its distance-matched partner. In subsequent sections, all mentions of ON or OFF medication refer to these state pairs unless mentioned otherwise.

We used the subject-level projection results obtained during IntraMed analysis to perform InterMed analyses. We performed two-sided independent sample t-tests between the matched states to compare the coherence, which was calculated between different regions of interest (see Dataset preparation). We grouped individual atlas regions into canonical cortical regions like frontal, sensorimotor, parietal, visual, medial PFC (prefrontal cortex), and STN contacts. For example, in the beta band, STN (contacts)–sensorimotor coherence in the OFF condition was compared to the STN (contacts)–sensorimotor coherence in the ON condition. The p-values obtained were corrected for multiple comparisons for a total number of possible combinations.

## Temporal properties of HMM states

To test for changes in the temporal properties OFF versus ON medication, we compared the lifetimes, interval between visits, and FO for each state both within and across HMMs using two-way repeated measures ANOVA followed by post hoc tests. Lifetime/dwell time of a state refers to the

time spent by the neural activity in that state. Interval of visit was defined as the time between successive visits of the same state. Finally, the FO of a state was defined as the fraction of time spent in each state. Extremely short state visits might not reflect neural processes, hence we only used values that were greater than 100 ms for lifetime comparisons.

## Acknowledgements

EF gratefully acknowledges support by the Volkswagen Foundation (Lichtenberg program 89387). Computational support and infrastructure was provided by the 'Centre for Information and Media Technology' (ZIM) at the University of Düsseldorf (Germany). We would like to thank Johannes Pfeifer for his valuable feedback on the manuscript.

## Additional information

### Competing interests

Alfons Schnitzler: has been serving as a consultant for Medtronic Inc, Boston Scientific, St. Jude Medical, Grünenthal, and has received lecture fees from Abbvie, Boston Scientific, St. Jude Medical, Medtronic Inc, UCB. The other authors declare that no competing interests exist.

### Funding

| Funder | Grant reference number | Author |
|---|---|---|
| Volkswagen Foundation | 89387 | Esther Florin |

The funders had no role in study design, data collection and interpretation, or the decision to submit the work for publication.

### Author contributions

Abhinav Sharma, Conceptualization, Software, Formal analysis, Investigation, Visualization, Methodology, Writing - original draft; Diego Vidaurre, Software, Validation, Methodology, Writing - review and editing; Jan Vesper, Alfons Schnitzler, Resources, Investigation, Writing - review and editing; Esther Florin, Conceptualization, Resources, Supervision, Funding acquisition, Validation, Investigation, Methodology, Project administration, Writing - review and editing

### Author ORCIDs

Abhinav Sharma ⓘ https://orcid.org/0000-0002-9296-3386
Diego Vidaurre ⓘ http://orcid.org/0000-0002-9650-2229
Esther Florin ⓘ https://orcid.org/0000-0001-8276-2508

### Ethics

Human subjects: The study was approved by the local ethics committee (study number 5608R) and conducted in accordance with the Declaration of Helsinki. Informed consent and consent to publish the results was obtained.

### Decision letter and Author response

Decision letter https://doi.org/10.7554/eLife.66057.sa1
Author response https://doi.org/10.7554/eLife.66057.sa2

## Additional files

### Supplementary files

- Transparent reporting form

## Data availability

We have made the code to produce the results and generate the figures available on Github: https://github.com/saltwater-tensor/HMM_pipeline (copy archived athttps://archive.softwareheritage.org/swh:1:rev:277a6a0ff21ff6885815c934255f953a97e16e98). However, the raw data cannot be made publicly available due to the European and German data privacy laws. When signing the informed consent forms, our patients consented to using their data for research purposes, but they did not sign a form stating that their data can be shared publicly, even in anonymized form. In addition, the MRIs of their heads and brains might make them identifiable. Hence, posting them within a public repository is not possible. The raw data can be requested from the corresponding author for replication of the current results and will then be shared in an anonymized way. We are providing the intermediate Matlab data underlying the figures with our submission.

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
