## [Decision Letter]

**Acceptance summary:**

In this manuscript, Sharma et al., investigated the effect of dopamine administration on oscillatory whole-brain networks by means of simultaneous local field potential recordings from the subthalamic nucleus and whole-brain magnetoencephalography recordings in seventeen patients with Parkinson's disease. A key feature is the combined use of invasive and noninvasive recordings and to investigate network changes by employing a hidden markov model. They identified three physiologically interpretable spectral connectivity patterns and found that cortico-cortical, cortico-STN, and STN-STN networks were differentially modulated by dopaminergic medication. These findings provide new insights regarding the mechanisms by which dopamine and medications alter cortico-basal ganglia dynamics, and open up new directions for studying their functions.

**Decision letter after peer review:**

Thank you for submitting your article "Differential dopaminergic modulation of spontaneous cortico-subthalamic activity in Parkinson's disease" for consideration by *eLife*. Your article has been reviewed by 3 peer reviewers, and the evaluation has been overseen by a Reviewing Editor and Michael Frank as the Senior Editor. The following individuals involved in review of your submission have agreed to reveal their identity: Kelly Bijanki (Reviewer #1); Muthuraman Muthuraman (Reviewer #2).

Essential Revisions:

1. I believe one strong concern is the first one raised by reviewer #1 in that it is important to differentiate neural activity differences in the ON medication state in whether they may originate from the effects of dopamine on the brain or the fact that the patients are moving more. Is there any possibility to address this issue? E.g. were motion parameters analysed to some degree? Does archival data exist that could be used to differentiate states at rest vs. states with movement (e.g. finger tapping)? If not this limitation should be prominently discussed.

2. While I personally think the concern of rev. 3 with contamination of STN signals by EEG is not of major concern (and the method has been established with reproducible results e.g. based on the Litvak / Hirschmann / Neumann works). Still, electrode localization could help to at least assure all leads were properly placed (or if not results would remain stable if the ones outside STN would be discarded)

3. I agree with rev. #2 that presentation can be improved. While circular connection plots are informative, it could be helpful to see to results in Figures 2-4 mapped to a brain (or sensor space). In general, I think the Results section could be stratified and presentation optimized.

*Reviewer #1 (Recommendations for the authors):*

Below is a list of concerns upon reviewing the manuscript, with suggestions for improvement.

1. The style of writing in the introduction is a little light on examination of the prior literature and motivation for the study. Starting at line 79, it transitions into a preview of the methods section listing of the analyses the researchers undertook and less like a true introduction setting the scene for the experiments.

2. This reviewer was surprised by the use of the word "communication", communication state, comms, etc. Perhaps this is a term of art in MEG (not this reviewer's area of expertise) but defining an MEG feature as indicative of communication seems presumptuous and merits at least brief discussion in the manuscript. Please add more background on MEG analysis and interpretation, especially in PD, in the introduction section.

3. This reviewer was unclear on why increased "communication" in the medial OFC in δ and theta was interpreted as a pathological state indicating deteriorated frontal executive function. Given that the authors provide no evidence of poor executive function in the patients studied, the authors must at least provide evidence from other studies linking this feature with impaired executive function.

4. Authors further reported that increased DA (L-DOPA administration) caused β activity to switch from STN-mediated motor network to a frontoparietal mediated one. The authors provide somewhat impoverished anatomical detail about the differences between the observed STN-mediated motor network and the known pathological β activity in STN in PD.

5. Last, authors report that DA didn't modify locally-originating STN oscillations in PD, but the detail on how they define locally-originating is unclear.

6. On line 86 the authors identify prior research limited by its investigation of specific connectivity pairs, whereas on line 104 the authors report on connectivity pairs in the current study.

7. The authors use the term "oscillations" without properly defining it in terms of neural activity and they have not addressed the analytical approach they use to confirm the activity measured on LFP or MEG reflects oscillatory activity vs. bursting activity or other neural activity types.

8. Authors need to acknowledge the role of DBS in PD therapy earlier in the study and identify that as the means by which they have access to simultaneous LFP recording. They don't even define the acronym "DBS" at its first use.

9. Authors could be more clear in lines 90-97 to define what they mean by "disruptive, "physiologically restorative", and "limited".

10. Authors repeatedly state their method allows them to delineate between pathological and physiological connectivity, but they don't explain how dynamical systems and discrete-state stochasticity support that goal. Lines 106 and 111.

11. Authors must address differences in neural activity other than DA-mediated changes in the on and off medication state. For example, when patients are off med, their UPDRS scores are elevated – do they not have movements that would pick up extra activations in MEG during "rest"? Is it possible to do a true "resting state" in active PD? At minimum, this concern must be discussed in the manuscript.

12. Figure 1: this reviewer would like to see the Y axis indicating amplitude or power, rather than arbitrary units from NNMF output. This figure also begs the question why the NMMF categorized 10-20hz oscillations as δ/theta (bump in blue line in "ON medication" and above 30Hz as α (red line, same plot)). To this reviewer, it appears as though the NMMF factorization worked effectively in the "OFF medication" data set but failed in the "ON medication" data set.

13. Patient sample appears to under-represent female patients. Is there a relationship between sex and DA metabolism/uptake/MEG markers? Perhaps simply acknowledge the imbalance in the manuscript.

14. LFP recordings were sampled from externalized DBS leads using the St. Jude directional 6172 lead except in one patient who had the Boston Scientific leads (which I think is fine to include in analysis together provided surface areas of contacts are identical). The geometry of the segmented leads needs to be addressed relative to sensitivity for various frequencies of oscillatory activity. Segmented leads have very low surface area and may require ultra-high sampling rates (30 kHz and higher) to resolve oscillatory activity.

*Reviewer #2 (Recommendations for the authors):*

General remarks:

1. A visualization of the state time series would be a nice addition to see the dynamics of the network. Including the probabilities would further give an impression of "how clear the states are", i.e. were they exclusive or were there some intervals where 2 states were rather active at the same time.

2. The color coding in table 1 (4x orange) does not correspond to the ring figures (2x orange/hemisphere). Both areas 17 and 18 are termed "medial orbitofrontal".

Results

As stated in the public review, there are several points concerning the presentation of the results:

3. Line 180 Supporting the dopamine overdose hypothesis in PD, we identified a δ/theta oscillatory network involving lateral and medial orbitofrontal cortical regions.

– Figure 2 also shows the inter-hemispheric connectivity of the pars orbitalis. What is the reason to not mention this in the main text?

4. Line 259: Furthermore, ON medication in the α band only the connectivity between temporal and parietal cortical regions and the STN was preserved (Figure 3C α; p < 0.05), consistent with previous findings (Litvak et al. 2011). In contrast, in the β band only STN-medial orbitofrontal connectivity remained intact (p < 0.05, Figure 3C β).

– Similar to the comment on line 180, the figure also shows α connectivity between the STN and medial orbitofrontal cortex.

5. Line 269: Finally, ON medication, a sensorimotor-frontoparietal network emerged (p < 0.05, Figure 3C β) where sensorimotor, frontal, and parietal regions were no longer connected to the STN, but instead directly communicated with each other in the β band.

– Similar to the comment on line 180: is there a reason to exclude the connection from the somatomotor cortex to the caudal middle frontal (e.g. figure 3c, β: 13r-16r)?

6. 256: Importantly, coherence OFF medication was significantly larger than ON medication between STN and sensorimotor, STN and temporal and STN and frontal cortices (p < 0.05 for all connections, Figure 3B α and β).

– The figure does not these results for STN and sensorimotor cortex

7. 263: Most previous PD studies report a decrease in the motor-STN coherence ON medication in the β band (Hammond, Bergman, and Brown 2007; Litvak et al. 2011; Hirschmann et al. 2013; Little et al. 2013; Marinelli et al. 2017) but do not indicate any changes that the sensorimotor regions might experience at the whole-brain level. In the communication state, OFF medication, STN-pre-motor (sensory), STN-frontal, and STN-parietal connectivity was present (p < 0.05, Figure 3A α and β). STN-cortical coherence was then significantly reduced ON compared to OFF medication (p < 0.05, Figure 3B α and β).

– The first sentence introduces cortico-cortical connections, which is then, however, followed by STN-cortical results. This is then followed by results on the sensorimotor-frontoparietal network, which indeed is cortico-cortical. However, later in the manuscript, the following statement suggests that the cortico-cortical network differs from the sensorimotor-frontoparietal network: "Still, significant connectivity was selectively preserved in a spectrally-specific manner ON medication both at the corticocortical level and the cortical-STN level. Furthermore, a sensorimotor-frontoparietal network emerged ON medication" (line282). It is not clear what exactly resembles the cortico-cortical network, and where the results are presented.

On the results of temporal properties:

8. Line 324: Previous research has shown that ON medication, spectrally-specific cortico-STN connectivity remains preserved in PD compared to OFF medication (Litvak et al. 2011; Hirschmann et al. 2013). This indicates the existence of functionally relevant cortico-STN loops. A decrease in coherence between the cortex and the STN has also been observed ON medication (George et al. 2013), which was correlated with improved motor functions in PD. All the connectivity effects were observed in our results for the communication state. Furthermore, in the communication state, we showed the existence of a frontoparietal sensorimotor 331 network in the β band ON medication. Recent evidence indicates that with the loss of dopamine and the start of PD symptoms, δ/theta oscillations emerge within the basal ganglia (Whalen et al. 2020). In line with these findings, STN-STN δ/theta oscillations in the local state were reduced ON medication.

– I have the impression that this paragraph is not in the right position. It repeats and even discusses the previously mentioned results and jumps from "frontoparietal sensorimotor networks" to "STN-STN δ/theta" findings.

Further, it is not clear if "All the connectivity effects were observed in our results for the communication state" relates to the citations above, especially since other states are shown to have effects on connectivity as well.

9. Additionally, the main text should provide t-scores and degrees of freedom, not only p-values. The comprehension of the results could be improved by displaying significantly different comparisons in a clear way in Figure 5 (e.g. provide * for p<0.05). However, what was the rationale for testing the medication condition and states with different t-tests? Would a 2way- repeated measures ANOVA not be more appropriate?

10. Line 353: The lifetimes for both the local and communication state were significantly increased by medication (ON > OFF: local, p < 0.01; comms, p {less than or equal to} 0.01).

and line 360: Both the local and communication state tended to last longer ON medication.

– This seems to be a repetition. Also, it is not clear what "tended to last longer" means? Is there a significant difference or a trend?

Methods

11. Line 676: Since NNMF does not guarantee a unique solution, we performed multiple instances of the factorization.

– The authors should state the exact number of repetitions. How was this number decided?*Reviewer #3 (Recommendations for the authors):*

1. The Authors could examine the effect of the dopaminergic medication (ON and OFF) on the recurrent oscillatory patterns of transient network connectivity within and between the cortex and the STN, as a function of the duration of the disease and/or the DRT history.

2. Re the contamination of the STN signal by volume conducted signals from the cortex. Page 20 (lines 584-586) "To correct for volume conduction in the signal, symmetric orthogonalisation (Colclough et al., 2015) was applied to each subject's resulting cortical time series matrix".

Have the authors applied this correction to the STN LFPs? If the authors cannot rule out the possibility of a contamination of STN LFPs by volume conducted signals from the cortex and cannot guarantee that the signal recorded in the STN only reflected local STN activity they should at least interpret their results with caution and discuss this limitation in their discussion.

3. The authors could make the paper more reader-friendly. In particular, could classical spectral and coherence analyses be performed to visualize the characteristics of the oscillatory activity and neural synchronization of the EMG and LFP signals in the recurrent oscillatory patterns of transient network connectivity both ON and OFF medication?

4. In this study, the authors could reconstruct the trajectories of the DBS leads within the STN (using for example the open source LeadDBS program?).

---

## [Author Response]

Essential Revisions:1. I believe one strong concern is the first one raised by reviewer #1 in that it is important to differentiate neural activity differences in the ON medication state in whether they may originate from the effects of dopamine on the brain or the fact that the patients are moving more. Is there any possibility to address this issue? E.g. were motion parameters analysed to some degree? Does archival data exist that could be used to differentiate states at rest vs. states with movement (e.g. finger tapping)? If not this limitation should be prominently discussed.

Regarding the higher movement ON medication during the resting condition, it would mainly be due to hyperkinesia. None of our patients did experience hyperkinesia during the recording. Unfortunately, we do not have actual movement data for our patients during the MEG recording.

Instead, we investigated the UPDRS 3 sub-scores for tremor and akinesia. According to this analysis, one patient was tremor-dominant and one of mixed type. As tremor causes additional movement in the OFF medication condition, we excluded these two patients and recalculated the HMM. ON medication results for all HMM states remained the same. OFF medication results for the Ctx-Ctx and STN-STN state remained the same as well. The Ctx-STN state OFF medication was split into two states: Sensorimotor-STN connectivity was captured in one state and all other types of Ctx-STN connections were captured in another state (see Author response image 1 and Author response image 2). The important point in the context of our study is that these different solutions preserve the biological conclusions: both with and without the two subjects a stable covariance matrix entailing sensorimotor-STN connectivity was determined, which is the main finding for the Ctx-STN state OFF medication.

**Author response image 1. respfig1:** States obtained after removing one tremor dominant and one mixed type patient from analysis.

**Author response image 2. respfig2:** States obtained after removing one tremor dominant and one mixed type patient from analysis. Panel C shows the split OFF medication cortico-STN state. Most of the cortico-STN connectivity is captured by the state shown in the top row (Figure 1 C OFF). Only the motor-STN connectivity in the α and β band (along with a medial frontal-STN connection in the α band) is captured separately by the states labeled “OFF Split” (Figure 1 C OFF SPLIT).

We therefore discuss this issue now within the limitation section (page 13):

“Both motor impairment and motor improvement can cause movement during the resting state in PD. While such movement is a deviation from a resting state in healthy subjects, such movements are part of the disease and occur unwillingly. Therefore, such movements can arguably be considered part of the resting state of Parkinson’s disease. None of the patients in our cohort experienced hyperkinesia during the recording. All patients except for two were of the akinetic-rigid subtype. We verified that tremor movement is not driving our results. Recalculating the HMM states without these 2 subjects, even though it slightly changed some particular aspects of the HMM solution, did not materially affect the conclusions.”

2) While I personally think the concern of rev. 3 with contamination of STN signals by EEG is not of major concern (and the method has been established with reproducible results e.g. based on the Litvak / Hirschmann / Neumann works). Still, electrode localization could help to at least assure all leads were properly placed (or if not results would remain stable if the ones outside STN would be discarded)

We thank the review editor for this suggestion. We have reconstructed the electrode location with Lead-DBS (Horn et al., 2019). All electrodes were properly placed within the STN – except for one (see Author response image 3 and Figure 7 in the manuscript). To exclude that our results were driven by outlier, we reanalysed our data without this patient. No apparent change in the overall connectivity pattern was observed (Author response image 4 and Author response image 5). In addition, we now discuss the potential of volume conduction of signals from the cortex to the STN as a potential limitation (page 13-14): “Lastly, we recorded LFPs from within the STN –an established recording procedure during the implantation of DBS electrodes in various neurological and psychiatric diseases. Although for Parkinson patients results on β and tremor activity within the STN have been reproduced by different groups (Reck et al., 2010, Litvak et al., 2011, Florin et al., 2013, Hirschmann et al., 2013, Neumann et al., 2016), it is still not fully clear whether these LFP signals are contaminated by volume-conducted cortical activity. However, while volume conduction seems to be a larger problem in rodents even after re-referencing the LFP signal (Lalla et al., 2017), the same was not found in humans (Marmor et al., 2017).”

**Author response image 3. respfig3:** Lead DBS reconstruction of the location of electrodes in the STN for different subjects. The red electrodes have not been placed properly in the STN. The contacts marked in red represent the directional contacts from which the data was used for analysis.

**Author response image 4. respfig4:** HMM states obtained after running the analysis without the subject with the electrode outside the STN.

**Author response image 5. respfig5:** HMM states obtained after running the analysis without the subject with the electrode outside the STN.

3) I agree with rev. #2 that presentation can be improved. While circular connection plots are informative, it could be helpful to see to results in Figures 2-4 mapped to a brain (or sensor space). In general, I think the Results section could be stratified and presentation optimized.

We have stratified the Results sections according to the suggestion of reviewer 2. In addition, we are now providing glass-brain views of the connections in figures 2-4.

Reviewer #1 (Recommendations for the authors):Below is a list of concerns upon reviewing the manuscript, with suggestions for improvement.1. The style of writing in the introduction is a little light on examination of the prior literature and motivation for the study. Starting at line 79, it transitions into a preview of the methods section listing of the analyses the researchers undertook and less like a true introduction setting the scene for the experiments.

We have changed the introduction and added more background literature (page 2-3):

“Dopamine is a widespread neuromodulator in the brain (Gershman and Uchida 2019), raising the question of whether each medication-induced change restores physiological oscillatory networks. In particular, dopaminergic medication is known to produce cognitive side effects in PD patients (Voon et al., 2009). According to the dopamine overdose hypothesis, a reason for these effects is the presence of excess dopamine in brain regions not affected in PD (MacDonald et al., 2011; MacDonald and Monchi, 2011). Previous task-based and neuroimaging studies in PD demonstrated frontal cognitive impairment due to dopaminergic medication (Cools et al., 2002; Ray and Strafella, 2010; MacDonald et al., 2011).

Using resting-state whole-brain MEG analysis, network changes related to both motor and non-motor symptoms of PD have been described (Olde Dubbelink et al., 2013a, b). However, these studies could not account for simultaneous STN-STN or cortico–STN activity affecting these networks, which would require combined MEG/EEG–LFP recordings (Litvak et al., 2021). Such recordings are possible during the implantation of deep brain stimulation (DBS) electrodes, an accepted treatment in the later stages of PD (Volkmann et al., 2004; Deuschl et al., 2006, Kleiner-Fisman et al., 2006). Combined MEG-LFP studies in PD involving dopaminergic intervention report changes in β and α band connectivity between specific cortical regions and the STN (Litvak et al., 2011; Hirschmann et al., 2013; Oswal et al., 2016). Decreased cortex-STN coherence under dopaminergic medication (ON) correlates with improved motor functions in PD (George et al., 2013). STN–STN intra-hemispheric oscillations positively correlate to motor symptom severity in PD without dopaminergic medication (OFF), whereas dopamine-dependent nonlinear phase relationships exist between inter-hemispheric STN–STN activity (West et al., 2016). Crucially, previous studies could not rule out the influence of cortico–STN connectivity on these inter-hemispheric STN–STN interactions.”

2. This reviewer was surprised by the use of the word "communication", communication state, comms, etc. Perhaps this is a term of art in MEG (not this reviewer's area of expertise), but defining an MEG feature as indicative of communication seems presumptuous and merits at least brief discussion in the manuscript. Please add more background on MEG analysis and interpretation, especially in PD, in the introduction section.

We apologize for any confusion the loaded terminology may have caused. Following this and reviewer 2’s similar comment we renamed the states. Now the communication state is termed Cortico-STN state. In the previous version of the manuscript, we used the word “communication” to denote connectivity between regions based on spectral coherence. The terminology was based on the communication through coherence (CTC) hypothesis (Fries, 2005, 2015), but we think the new label is more descriptive. More background on MEG analysis in PD has been added to the manuscript (see previous reply). In addition, we added the following explanation to the introduction (page 3):

“We study whole-brain connectivity including the STN using spectral coherence as a proxy for communication based on the communication through coherence hypothesis (Fries, 2005, 2015). This will allow us to delineate differences in communication OFF and ON medication”

3. This reviewer was unclear on why increased "communication" in the medial OFC in δ and theta was interpreted as a pathological state indicating deteriorated frontal executive function. Given that the authors provide no evidence of poor executive function in the patients studied, the authors must at least provide evidence from other studies linking this feature with impaired executive function.

If we understand the comment correctly it refers to the statement in the abstract “Dopaminergic medication led to communication within the medial and orbitofrontal cortex in the delta/theta frequency range. This is in line with deteriorated frontal executive functioning as a side effect of dopamine treatment in Parkinson’s disease”

This statement is based on the dopamine overdose hypothesis reported in the Parkinson’s disease (PD) literature (Cools, 2001; Kelly et al., 2009; MacDonald and Monchi, 2011; Vaillancourt et al., 2013). We have elaborated upon the dopamine overdose hypothesis in the discussion on page 10. In short, dopaminergic neurons are primarily lost from the substantia nigra in PD, which causes a higher dopamine depletion in the dorsal striatal circuitry than within the ventral striatal circuits (Kelly et al., 2009; MacDonald and Monchi, 2011). Thus, dopaminergic medication to treat the PD motor symptoms leads to increased dopamine levels in the ventral striatal circuits including frontal cortical activity, which can potentially explain the cognitive deficits observed in PD (Shohamy et al., 2005; George et al., 2013). We adjusted the abstract to read:

“Dopaminergic medication led to coherence within the medial and orbitofrontal cortex in the delta/theta frequency range. This is in line with known side effects of dopamine treatment such as deteriorated executive functions in Parkinson’s disease.”

4. Authors further reported that increased DA (L-DOPA administration) caused β activity to switch from STN-mediated motor network to a frontoparietal mediated one. The authors provide somewhat impoverished anatomical detail about the differences between the observed STN-mediated motor network and the known pathological β activity in STN in PD.

Prior research has indicated that β activity within the dorsolateral STN is related to the pathology of PD (Marreiros et al., 2013; van Wijk et al., 2016; West et al., 2018). To provide more anatomical detail on the electrode location we reconstructed the electrode location using Lead-DBS (Horn et al., 2019). Except for one electrode, all electrode contacts used in the analysis were located within the dorsolateral STN. We have added a figure with the electrode locations to the manuscript (see manuscript figure 7 and Author response image 3).

Consistent with previous studies in the β frequency band OFF medication, we found STN-sensorimotor (Hirschmann et al. 2011), STN-parietal (inferior parietal) (Litvak et al., 2011), STN-frontal (insular cortex, pars orbitalis, pars opercularis and lateral orbitofrontal) (Litvak et al., 2011), and STN-medial prefrontal (medial orbitofrontal) (Litvak et al., 2011) coherence in our Ctx-STN state results. ON medication sensorimotor regions were coherent with parietal (para central) and frontal (superior frontal)/ medial prefrontal (caudal middle frontal) regions in the β band. Previous research has not reported the emergence of such a coherent network ON medication. But consistent with previous research, we found that sensorimotor-STN coherence was reduced ON medication (Hirschmann et al., 2013).

We provide details in the discussion on page 11:

“Previous studies have reported STN-sensorimotor (Hirschmann et al., 2011, Litvak et. al., 2011), STN-parietal, and STN-frontal (Litvak et al., 2011) coherence in the β band OFF medication. Consistent with previous studies STN-sensorimotor, STN-parietal (inferior parietal), STN-frontal (Insular cortex, pars orbitalis, pars opercularis and lateral orbitofrontal), and STN-medial prefrontal (medial orbitofrontal) coherence emerged in the Ctx-STN state. In contrast, ON medication sensorimotor regions were coherent with parietal (para central) and frontal (superior frontal)/ medial prefrontal (caudal middle frontal) regions in the β frequency range. Previous research has not reported the emergence of such a coherent fronto-parietal-motor network ON medication. But consistent with previous research (Hirschmann et al., 2013), sensorimotor-STN coherence was reduced ON compared to OFF medication.”

5. Last, authors report that DA didn't modify locally-originating STN oscillations in PD, but the detail on how they define locally-originating is unclear.

We apologize for not more clearly defining this term. Because only STN-STN coherence/connectivity is altered under medication in the “local state”, we termed this state “local state”. But we agree the term is confusing and changed it to “STN-STN state”. Unfortunately, identifying whether the STN oscillations are indeed originating locally within basal ganglia circuits would require a higher coverage of invasive recording locations as well as intracellular recordings. We changed the abstract to:

“In contrast, dopamine did not modify local STN-STN coherence in PD.”

6. On line 86 the authors identify prior research limited by its investigation of specific connectivity pairs, whereas on line 104 the authors report on connectivity pairs in the current study.

We are sorry for this unclear description. The line stating “Each HMM state was characterised by coherence calculated between different pair of regions” was supposed to intuitively explain what an HMM state is: a multidimensional, time-delay embedded covariance matrix capturing frequency-specific connectivity between different brain regions. Each HMM state can subsequently be analysed in a spectrally-specific manner. To eliminate this seemingly contradictory statement we changed this sentence to (page 4): “Each HMM state itself is a multidimensional, time-delay embedded covariance matrix across the whole brain, containing information about cross-regional coherence and power in the frequency domain.”

Analysing whole brain connectivity instead of investigating pre-specified bilateral connectivity pairs is an important difference to previous studies. Thus, we investigated coherence between multiple regions and report only those connectivity pairs that significantly stand out from all the possible functional connections.

7. The authors use the term "oscillations" without properly defining it in terms of neural activity and they have not addressed the analytical approach they use to confirm the activity measured on LFP or MEG reflects oscillatory activity vs. bursting activity or other neural activity types.

For the purpose of our paper, oscillatory activity or oscillations refers to recurrent but transient frequency–specific patterns of network activity. Because our analysis pipeline relies on an HMM, the oscillatory connectivity under consideration is neither exclusively made up of sustained rhythmic activity nor neural bursting, and it can be a mix of both (van Ede et al., 2018; Quinn et al., 2019). We have added this information as a footnote to the paper on page 2.

“For the purposes of our paper we refer to oscillatory activity or oscillations as recurrent but transient frequency–specific patterns of network activity, even though the underlying patterns can be composed of either sustained rhythmic activity, neural bursting, or both (Quinn et al., 2019). Disambiguating the exact nature of these patterns is, however, beyond the scope of the present work.”

8. Authors need to acknowledge the role of DBS in PD therapy earlier in the study and identify that as the means by which they have access to simultaneous LFP recording. They don't even define the acronym "DBS" at its first use.

Thank you for pointing this out. We now define DBS at its first use and added the information on DBS surgery as a further treatment option to the introduction. We added on page 2:

“However, these studies could not account for simultaneous STN-STN or cortico–STN activity affecting these networks, which would require combined MEG/EEG–LFP recordings (Litvak et al. 2021). Such recordings are possible during the implantation of deep brain stimulation (DBS) electrodes, an accepted treatment in the later stages of PD (Volkmann et al., 2004; Deuschl et al., 2006, Kleiner-Fisman et al., 2006)”

9. Authors could be more clear in lines 90-97 to define what they mean by "disruptive, "physiologically restorative", and "limited".

We have changed those lines to make the reported findings more accessible. We used “disruptive” to denote that connectivity changes due to dopamine were related to the cognitive side effects reported in literature. “Physiologically restorative” was supposed to refer to connectivity changes on medication that brought connectivity closer to normal physiological connectivity as previously reported in PD. Finally, “limited” referred to the fact that there were only changes in STN-STN coherence in that state.

The relevant passage now reads (page 3):

“For the cortico–cortical network medication led to additional connections that can be linked to the side effects of dopamine. At the same time, dopamine changed the cortico-STN network towards a pattern more closely resembling physiological connectivity as reported in the PD literature. Within the third network, dopamine only had an influence on local STN-STN coherence. “

10. Authors repeatedly state their method allows them to delineate between pathological and physiological connectivity, but they don't explain how dynamical systems and discrete-state stochasticity support that goal. Lines 106 and 111.

To recapitulate, the HMM divides a continuous time series into discrete states. Each state is a time-delay embedded covariance matrix reflecting the underlying connectivity between brain regions as well as the specific temporal dynamics in the data when such state is active. See Packard et al., (1980) for details about how a time-delay embedding characterises a linear dynamical system.

Please note that the HMM was used as a data-driven, descriptive approach without explicitly assuming any a-priori relationship with pathological or physiological states. The relation between biology and the HMM states, thus, purely emerged from the data; i.e. is empirical. What we claim in this work is simply that the features captured by the HMM hold some relation with the physiology even though the estimation of the HMM was completely unsupervised (i.e. blind to the studied conditions). We have added this point also to the limitations of the study on page 18 and the following to the introduction to guide the reader more intuitively (page 4):

“To allow the system to dynamically evolve, we use time delay embedding. Theoretically, delay embedding can reveal the state space of the underlying dynamical system (Packard et al., 1980). Thus, by delay-embedding PD time series OFF and ON medication we uncover the differential effects of a neurotransmitter such as dopamine on underlying whole brain connectivity.”

11. Authors must address differences in neural activity other than DA-mediated changes in the on and off medication state. For example, when patients are off med, their UPDRS scores are elevated – do they not have movements that would pick up extra activations in MEG during "rest"? Is it possible to do a true "resting state" in active PD? At minimum this concern must be discussed in the manuscript.

We agree that Parkinson’s disease can lead to unwanted movements such as tremor as well as hyperkinesias. This would of course be a deviation from a resting state in healthy subjects. However, such movements are part of the disease and occur unwillingly. The main tremor in Parkinson’s disease is a rest tremor and – as the name already suggests – it occurs while not doing anything. Therefore, such movements can arguably be considered part of the resting state of Parkinson’s disease. Resting state activity with and without medication is therefore still representative for changes in brain activity in Parkinson’s patients and indicative of alterations due to medication.

To further investigate the effect of movement in our patients, we subdivided the UPDRS part 3 score into tremor and non-tremor subscores. For the tremor subscore we took the mean of item 15 and 17 of the UPDRS, whereas for the non-tremor subscore items 1, 2, 3, 9, 10, 12, 13, and 14 were averaged. Following Spiegel et al., 2007, we classified patients as akinetic-rigid (non-tremor score at least twice the tremor score), tremor-dominant (tremor score at least twice as large as the non-tremor score), and mixed type (for the remaining scores). Of the 17 patients, 1 was tremor dominant and 1 was classified as mixed type (his/her non-tremor score was greater than tremor score). None of our patients exhibited hyperkinesias during the recording. To exclude that our results are driven by tremor-related movement, we re-ran the HMM without the tremor-dominant and the mixed-type patient (see Author response image 1 and Author response image 2).

ON medication results for all HMM states remained the same. OFF medication results for the Ctx-Ctx and STN-STN state remained the same as well. The Ctx-STN state OFF medication was split into two states: Sensorimotor-STN connectivity was captured in one state and all other types of Ctx-STN connections were captured in another state (see Figure 1 response letter). The important point is that the biological conclusions stand across these solutions. Regardless, both with and without the two subjects a stable covariance matrix entailing sensorimotor-STN connectivity was determined, which is the main finding for the Ctx-STN state OFF medication.

We therefore discuss this issue now within the limitation section (page 13):

“Both motor impairment and motor improvement can cause movement during the resting state in PD. While such movement is a deviation from a resting state in healthy subjects, such movements are part of the disease and occur unwillingly. Therefore, such movements can arguably be considered part of the resting state of Parkinson’s disease. None of the patients in our cohort experienced hyperkinesia during the recording. All patients except for two were of the akinetic-rigid subtype. We verified that tremor movement is not driving our results. Recalculating the HMM states without these 2 subjects, even though it slightly changed some particular aspects of the HMM solution did not materially affect the conclusions.”

12. Figure 1: this reviewer would like to see the Y axis indicating amplitude or power, rather than arbitrary units from NNMF output. This figure also begs the question why the NMMF categorized 10-20hz oscillations as δ/theta (bump in blue line in "ON medication" and above 30Hz as α (red line, same plot). To this reviewer, it appears as though the NMMF factorization worked effectively in the "OFF medication" data set but failed in the "ON medication" data set.

We apologise for the misunderstanding. The graphs in manuscript figure 1 do not reflect power, but the spectral templates derived from the NNMF method; that is, these values are the actual NNMF weights obtained from the NNMF estimation. These templates were in fact derived from coherence measures (as detailed in Vidaurre et al., 2018), which is a unit less measure. Therefore, these NNMF weights also don’t have a unit. We have clarified this in methods section titled Intra-medication analysis on page 20.

Please also note that NNMF is a data-driven technique and it is therefore natural that the spectral templates do not exactly match the canonical frequency bands. In particular, it is common that residual “bumps” emerge besides the main characteristic frequency (for example, as seen in the “α” band ON medication). However, given that all the results are statistically thresholded at a p-value of 0.05, any connections purely arising from the frequency bumps at 10-20 /20-40Hz Hz for the δ / α mode would be discarded as background noise, because the weights for the bumps observed in these modes are lower than the peak values in the respective mode. The analysis and selection of the frequency modes follows the analysis pipeline described and validated by Vidaurre et al., (2018).

13. Patient sample appears to under-represent female patients. Is there a relationship between sex and DA metabolism/uptake/MEG markers? Perhaps simply acknowledge the imbalance in the manuscript.

We now acknowledge the sex imbalance in the discussion, caused by more male patients having been admitted for DBS surgery. To the best of our knowledge, there is no previous literature reporting a sex difference in MEG markers. Previous studies have found the density of dopamine transporters to be higher in females than in males, which is in line with findings from animal studies (Lavalaye et al., 2000). On the other hand, previous work has not found any significant differences between men and women in the type and dose of dopaminergic medication used in PD (Umeh et al., 2014).

We have added the following paragraph to the limitations of study on page 13:

“In addition, we only recorded 4 female patients because during the study period fewer female patients underwent DBS surgery at our centre. To the best of our knowledge, there is no previous literature reporting a sex difference in MEG markers or the prescribed dopaminergic medication (Umeh et al., 2014).”

14. LFP recordings were sampled from externalized DBS leads using the St. Jude directional 6172 lead except in one patient who had the Boston Scientific leads (which I think is fine to include in analysis together provided surface areas of contacts are identical). The geometry of the segmented leads needs to be addressed relative to sensitivity for various frequencies of oscillatory activity. Segmented leads have very low surface area and may require ultra-high sampling rates (30 kHz and higher) to resolve oscillatory activity.

The surface area of the directional contacts is almost the same for the two systems: In both cases, the contact is 1.5mm long, but for the St. Jude electrode the diameter is 1.29mm, while for the Boston scientific lead it is 1.3mm. Our data was acquired at 2.4 KHz. Other studies using directional electrodes for LFP recordings employed 1Khz (Telkes et al., 2020) and 24 KHz down-sampled to 375 Hz for analysis (Nguyen et al., 2020) and were able to resolve oscillatory activity in the STN. Thus we are confident the sampling rate is sufficient. We now discuss this issue in limitations of the manuscript on page 14.

“Moreover, we used directional contacts, which have a smaller surface area than the classical ring contacts. Based on the available literature, our sampling rate was high enough to resolve oscillatory activity in the STN (Telkes et al., 2020; Nguyen et al., 2020).”

Reviewer #2 (Recommendations for the authors):General remarks:1. A visualization of the state time series would be a nice addition to see the dynamics of the network. Including the probabilities would further give an impression of "how clear the states are", i.e. were they exclusive or were there some intervals where 2 states were rather active at the same time.

We now provide a snapshot (out of the 6,425,750 time points) of the state probability time course as figure 5 in the manuscript. In addition, we provide the state time series for anyone to check and visualise.

2. The color coding in table 1 (4x orange) does not correspond to the ring figures (2x orange/hemisphere). Both areas 17 and 18 are termed "medial orbitofrontal".

Thanks – we have corrected the table.

ResultsAs stated in the public review, there are several points concerning the presentation of the results:3. Line 180 Supporting the dopamine overdose hypothesis in PD, we identified a δ/theta oscillatory network involving lateral and medial orbitofrontal cortical regions.– Figure 2 also shows the inter-hemispheric connectivity of the pars orbitalis. What is the reason to not mention this in the main text?

We thank the reviewer for pointing this out and we now also mention the missing inter-hemispheric connection in the text.

4. Line 259: Furthermore, ON medication in the α band only the connectivity between temporal and parietal cortical regions and the STN was preserved (Figure 3C α; p < 0.05), consistent with previous findings (Litvak et al. 2011). In contrast, in the β band only STN-medial orbitofrontal connectivity remained intact (p < 0.05, Figure 3C β).– Similar to the comment on line 180, the figure also shows α connectivity between the STN and medial orbitofrontal cortex.

We have now also added this connection to the text.

5. Line 269: Finally, ON medication, a sensorimotor-frontoparietal network emerged (p < 0.05, Figure 3C β) where sensorimotor, frontal, and parietal regions were no longer connected to the STN, but instead directly communicated with each other in the β band.– Similar to the comment on line 180: is there a reason to exclude the connection from the somatomotor cortex to the caudal middle frontal (e.g. figure 3c, β: 13r-16r)?

We added this information to the text.

6. 256: Importantly, coherence OFF medication was significantly larger than ON medication between STN and sensorimotor, STN and temporal and STN and frontal cortices (p < 0.05 for all connections, Figure 3B α and β).– The figure does not these results for STN and sensorimotor cortex

We apologize for the confusion. Panel B of Figures 2-4 is thresholded for a p-value of 0.01, whereas the p-value for the STN-sensorimotor comparison was 0.01398 (β) and 0.02826 (α). Hence, these results were not visible in Figure 3B. We have changed panel B of figures 2-4 to a consistent threshold of 0.05.

7. 263: Most previous PD studies report a decrease in the motor-STN coherence ON medication in the β band (Hammond, Bergman, and Brown 2007; Litvak et al. 2011; Hirschmann et al. 2013; Little et al. 2013; Marinelli et al. 2017) but do not indicate any changes that the sensorimotor regions might experience at the whole-brain level. In the communication state, OFF medication, STN-pre-motor (sensory), STN-frontal, and STN-parietal connectivity was present (p < 0.05, Figure 3A α and β). STN-cortical coherence was then significantly reduced ON compared to OFF medication (p < 0.05, Figure 3B α and β).– The first sentence introduces cortico-cortical connections, which is then, however, followed by STN-cortical results. This is then followed by results on the sensorimotor-frontoparietal network, which indeed is cortico-cortical. However, later in the manuscript, the following statement suggests that the cortico-cortical network differs from the sensorimotor-frontoparietal network: "Still, significant connectivity was selectively preserved in a spectrally-specific manner ON medication both at the corticocortical level and the cortical-STN level. Furthermore, a sensorimotor-frontoparietal network emerged ON medication" (line282). It is not clear what exactly resembles the cortico-cortical network, and where the results are presented.

We like to thank the reviewer for pointing this out. We have added that “cortico-cortical network” refers to the sensorimotor-frontoparietal network. The results are presented on page 6:

“Finally, ON medication, a sensorimotor–frontoparietal network emerged (p < 0.05, Figure 3C β), where sensorimotor, medial prefrontal, frontal, and parietal regions were no longer connected to the STN, but instead directly communicated with each other in the β band. Hence, there was a transition from STN-mediated sensorimotor connectivity to the cortex OFF medication to a more direct cortico–cortical connectivity ON medication.”

On the results of temporal properties:8. Line 324: Previous research has shown that ON medication, spectrally-specific cortico-STN connectivity remains preserved in PD compared to OFF medication (Litvak et al. 2011; Hirschmann et al. 2013). This indicates the existence of functionally relevant cortico-STN loops. A decrease in coherence between the cortex and the STN has also been observed ON medication (George et al. 2013), which was correlated with improved motor functions in PD. All the connectivity effects were observed in our results for the communication state. Furthermore, in the communication state, we showed the existence of a frontoparietal sensorimotor 331 network in the β band ON medication. Recent evidence indicates that with the loss of dopamine and the start of PD symptoms, δ/theta oscillations emerge within the basal ganglia (Whalen et al. 2020). In line with these findings, STN-STN δ/theta oscillations in the local state were reduced ON medication.– I have the impression that this paragraph is not in the right position. It repeats and even discusses the previously mentioned results and jumps from "frontoparietal sensorimotor networks" to "STN-STN δ/theta" findings.Further, it is not clear if "All the connectivity effects were observed in our results for the communication state" relates to the citations above, especially since other states are shown to have effects on connectivity as well.

We have removed the mentioned paragraph and now only introduce the reason for investigating the temporal properties (page 8).

9. Additionally, the main text should provide t-scores and degrees of freedom, not only p-values. The comprehension of the results could be improved by displaying significantly different comparisons in a clear way in Figure 5 (e.g. provide * for p<0.05). However, what was the rationale for testing the medication condition and states with different t-tests? Would a 2way- repeated measures ANOVA not be more appropriate?

We thank the reviewer for pointing this out. Indeed, a two-way repeated measures ANOVA is more appropriate. We now report the results of the ANOVA and of the post-hoc tests, including the relevant degrees of freedom. Manuscript figure 6 and text (page 8) was updated accordingly.

“Two-way repeated measures ANOVA on the temporal properties of the HMM states revealed an effect of HMM states on the fractional occupancy (F (2,96) = 10.49, p< 0.01), interval of visits (F (2, 221513) = 9783.13, p < 0.01), and lifetime (F (2, 214818) = 50.36, p< 0.01). There was no effect of medication (L-DOPA) on fractional occupancy (F (1, 96) = 2.00, p = 0.16) and lifetime (F (1,214818) = 0.15, p = 0.7026). Medication had a significant effect on the interval of visits (F (1,221513) = 4202.96, p < 0.01). Finally, we found an interaction between the HMM states and medication on the interval of visits (F (2,221513) = 1949.98, p< 0.01) and lifetime (F (2,214818) = 172.25, p< 0.01). But there was no interaction between HMM states and medication on fractional occupancy (F (2,96) = 0.54, p = 0.5855). “

10. Line 353: The lifetimes for both the local and communication state were significantly increased by medication (ON > OFF: local, p < 0.01; comms, p {less than or equal to} 0.01).and line 360: Both the local and communication state tended to last longer ON medication.– This seems to be a repetition. Also, it is not clear what "tended to last longer" means? Is there a significant difference or a trend?

We have removed those repetitions. “Tended to last longer” was meant to indicate that the lifetime of the communication and local states was significantly increased on medication. This part now reads (page 9):

“In summary, the cortico-cortical state was visited least often compared to the other two states both OFF and ON medication. The cortico-STN and STN-STN states showing physiologically-relevant spectral connectivity on the other hand lasted significantly longer ON medication. “

Methods11. Line 676: Since NNMF does not guarantee a unique solution, we performed multiple instances of the factorization.– The authors should state the exact number of repetitions. How was this number decided?

Every iteration of the NNMF algorithm produces a new solution from scratch. We repeated the NNMF algorithm until the obtained frequency modes converged to an approximation of the classical frequency bands (consider this to be a Bayesian prior). This approach follows Vidaurre et al., 2018. In practice, this required 4 iterations of the NNMF algorithm. We added this information to the manuscript, on page 20.

Reviewer #3 (Recommendations for the authors):1. The Authors could examine the effect of the dopaminergic medication (ON and OFF) on the recurrent oscillatory patterns of transient network connectivity within and between the cortex and the STN, as a function of the duration of the disease and/or the DRT history.

We would like to thank the reviewer for pointing this out. We regressed duration of disease (year of measurement – year of onset) on the temporal properties of the HMM states. We found no relationship between any of the temporal properties and disease duration. Similarly, we regressed levodopa equivalent dosage for each subject on the temporal properties and found no relationship. We now discuss this point in the manuscript (page 13):

“A further potential influencing factor might be the disease duration and the amount of dopamine patients are receiving. Both factors were not significantly related to the temporal properties of the states.”

2. Re the contamination of the STN signal by volume conducted signals from the cortex. Page 20 (lines 584-586) "To correct for volume conduction in the signal, symmetric orthogonalisation (Colclough et al. 2015) was applied to each subject's resulting cortical time series matrix".Have the authors applied this correction to the STN LFPs? If the authors cannot rule out the possibility of a contamination of STN LFPs by volume conducted signals from the cortex and cannot guarantee that the signal recorded in the STN only reflected local STN activity they should at least interpret their results with caution and discuss this limitation in their discussion.

We apologize for not stating this more clearly. We did not apply symmetric orthogonalisation to the STN signals. We have changed the relevant sentence to (page 17):

“To correct for volume conduction in the signal, symmetric orthogonalisation (Colclough et al. 2015) was applied to each subject’s resulting MEG cortical time series matrix.”

With respect to volume conduction, we appreciate this concern and thank the reviewer for bringing it up. Marmor et al., (2017) investigated this on humans and is therefore most closely related to our research. They find that re-referenced STN recordings are not contaminated by cortical signals. Furthermore, the data in Lalla et al., (2017) is based on recordings in rats, making a direct transfer to human STN recordings problematic due to the different brain sizes. Since we re-referenced our LFP signals as recommended in the Marmor paper, we think that contamination due to cortical signals is relatively minor; see Litvak et al., (2011), Hirschmann et al. (2013), and Neumann et al., (2016) for additional references supporting this. That being said, we now discuss this potential issue in the paper on page 13-14.

“Lastly, we recorded LFPs from within the STN –an established recording procedure during the implantation of DBS electrodes in various neurological and psychiatric diseases. Although for Parkinson patients results on β and tremor activity within the STN have been reproduced by different groups (Reck et al., 2010, Litvak et al., 2011, Florin et al., 2013, Hirschmann et al., 2013, Neumann et al., 2016), it is still not fully clear whether these LFP signals are contaminated by volume-conducted cortical activity. However, while volume conduction seems to be a larger problem in rodents even after re-referencing the LFP signal (Lalla et al., 2017), the same was not found in humans (Marmor et al., 2017).”

3. The authors could make the paper more reader-friendly. In particular, could classical spectral and coherence analyses be performed to visualize the characteristics of the oscillatory activity and neural synchronization of the EMG and LFP signals in the recurrent oscillatory patterns of transient network connectivity both ON and OFF medication?

We now explain our analysis more intuitively within the manuscript (see page 4). To aid intuition on how to interpret the result in light of the methods used, one can compare the analysis pipeline to a windowing approach. In a more standard approach, windows of different time length can be defined for different epochs within the time series and for each window coherence and connectivity can be determined. The difference in our approach is that we used an unsupervised learning algorithm to select windows of varying length based on recurring patterns of whole brain network activity. Within those defined windows we then determine the oscillatory properties via coherence and power – which is the same as one would do in a classical analysis. We have added an explanation of the concept of “oscillatory activity” within our framework to the introduction (page 2 footnote):

“For the purpose of our paper, we refer to oscillatory activity or oscillations as recurrent, but transient frequency–specific patterns of network activity, even though the underlying patterns can be composed of either sustained rhythmic activity, neural bursting, or both (Quinn et al. 2019).”

Moreover, we provide a more intuitive explanation of the analysis within the first section of the results (page 4):

“Using an HMM, we identified recurrent patterns of transient network connectivity between the cortex and the STN, which we henceforth refer to as an ‘HMM state’. In comparison to classic sliding-window analysis, an HMM solution can be thought of as a data-driven estimation of time windows of variable length (within which a particular HMM state was active): once we know the time windows when a particular state is active, we compute coherence between different pairs of regions for each of these recurrent states.”

4. In this study, the authors could reconstruct the trajectories of the DBS leads within the STN (using for example the open source LeadDBS program?).

We selected the electrode contacts based on intraoperative microelectrode recordings (for details, see page 16). The first directional recording height after the entry into the STN was selected to obtain the three directional LFP recordings from the respective hemisphere. This practice has been proven to improve target location (Kochanski et al., 2019; Krauss et al., 2021). The common target area for DBS surgery is the dorsolateral STN. To confirm that the electrodes were actually located within this part of the STN, we now reconstructed the DBS location with Lead-DBS (Horn et al., 2019). All electrodes – except for one – were located within the dorsolateral STN (see figure 7 of the manuscript). To exclude that our results were driven by outlier, we reanalysed our data without this patient. No change in the overall connectivity pattern was observed (see Author response image 4 and Author response image 5).